# Faster Repeated Evasion Attacks in Tree Ensembles

**Lorenzo Cascioli**
Department of Computer Science
KU Leuven
Leuven, Belgium
`lorenzo.cascioli@kuleuven.be`

**Laurens Devos**
Department of Computer Science
KU Leuven
Leuven, Belgium

**Ondrej Kuzelka**
Faculty of Electrical Engineering
Czech Technical University in Prague
Prague, Czech Republic

**Jesse Davis**
Department of Computer Science
KU Leuven
Leuven, Belgium

## Abstract

Tree ensembles are one of the most widely used model classes. However, these models are susceptible to adversarial examples, i.e., slightly perturbed examples that elicit a misprediction. There has been significant research on designing approaches to construct such examples for tree ensembles. But this is a computationally challenging problem that often must be solved a large number of times (e.g., for all examples in a training set). This is compounded by the fact that current approaches attempt to find such examples from scratch. In contrast, we exploit the fact that multiple similar problems are being solved. Specifically, our approach exploits the insight that adversarial examples for tree ensembles tend to perturb a consistent but relatively small set of features. We show that we can quickly identify this set of features and use this knowledge to speedup constructing adversarial examples.

## 1 Introduction

One of the most popular and widely used classes of models is tree ensembles which encompasses techniques such as gradient boosting [16] and random forests [3]. However, like other flexible model classes such as (deep) neural networks [28, 17], they are susceptible to evasion attacks [23]. That is, an adversary can craft an imperceptible perturbation that, when applied to an otherwise valid input example, elicits a misprediction by the ensemble. As an example, consider a bank that uses a learned model to assess whether to approve or deny loan applications. In this setting, an evasion attack could entail slightly altering a potential customer's data (e.g., adding one month to their work seniority) that results in the model making a different decision on the customer's application. The slightly modified customer record is an adversarial example. There is significant interest in reasoning about tree ensembles to both generate such adversarial examples [15, 36] and perform empirical robustness checking [23, 8, 10] where the goal is to determine how close the nearest adversarial example is.

Generating adversarial examples is an NP-hard problem [23], which has spurred the development of approximate techniques [8, 36, 10]. These methods exploit the structure of the trees to find adversarial examples faster, e.g., by using graph transformations [8] or discrete (heuristic) search [36, 10, 12]. Still, these techniques can be slow, particularly if there is a large number of attributes in the domain. This is compounded by the fact that one often wants to generate large sets of adversarial examples.

A weakness to existing approaches is that they ignore the fact that adversarial example generation is often a sequential task where multiple similar problems are being solved in a row. That is, one has access to a large number of "normal" examples each of which should be perturbed to elicit

a misprediction. Alas, existing approaches treat each considered example in isolation and solve the problem from scratch. However, there are likely regularities among the problems, meaning that the algorithms perform redundant work. If these regularities can be identified efficiently and this information can be exploited to guide the search for an adversarial example, then the run time performance of repeated adversarial example generation can be improved.

Studying these regularities in order to make adversarial example generation faster is an important problem. First, it advances our understanding of the nature of adversarial examples in tree ensembles and their generation methods. This might inspire improvements to generation methods, and in turn lead to better defense or detection methods. Second, model evaluation by verification [26, 29, 10] is quickly becoming important as machine learning is applied in sensitive application areas. Being able to efficiently generate adversarial examples is crucial for computing empirical robustness (e.g., [10]), adversarial accuracy (e.g., [32]), and for model hardening (e.g., [23]). Third, some scenarios exist where an attacker would want to perform a large scale evasion attack. For example, some DNS registries use models to flag new domain registrations as potentially malicious (e.g., for phishing, fake webshops) [27] and scammers likely need to register many such domains. Finally, techniques in the planning community for analyzing policy safety through predicate abstraction involve performing repeated verification queries on the same model [30, 31, 22].

We propose a novel approach that analyzes previously solved adversarial example generation tasks to inform the search for subsequent tasks. Our approach is based on the observation that for a fixed learned tree ensemble, adversarial examples tend to be generated by perturbing the same, relatively small set of features. We propose a theoretically grounded manner to quickly find this set of features. We then propose two novel strategies to use the identified features to guide the search for adversarial examples, one of which is guaranteed to produce an adversarial example if it exists. We apply our proposed approach to two different algorithms for generating adversarial examples [23, 10]. Empirically, our approaches result in speedups of up to 36x/21x and on average of 9x/4x ($\pm$ 8x/3x). The source code for the presented algorithms and all the experiments is publicly available at `https://github.com/lorenzocascioli/faster-repeated-evasion-tree-ensembles`.

## 2 Preliminaries

We briefly explain tree ensembles, evasion attacks, and the two adversarial generation methods used in the experiments. We assume a $d$-dimensional input space $\mathcal{X} \subseteq \mathbb{R}^d$ and binary output space $\mathcal{Y} = \{-1, 1\}$. We focus on binary classification because most existing methods for generating adversarial examples for tree ensembles are designed for this setting [1, 23, 10].

**Tree Ensembles**  Tree ensembles include algorithms such as (gradient) boosted decision trees (GBDTs) [16, 9] and random forests [3, 25]. A tree ensemble contains a number of trees and most implementations only learn binary trees. A binary tree $T$ contains two types of nodes. *Internal nodes* store references to a left and a right sub-tree, and a split condition on some attribute $f$ in the form of a less-than comparison $X_f < \tau$, where $\tau$ is the split value. *Leaf nodes* have no children and only contain an output value. Each tree starts with a *root node*, the only one without a parent.

Given an example $x$, an individual tree is evaluated recursively starting from the *root node*. In each internal node, the split condition is applied and if it is satisfied, then the example is sorted to the left subtree and if not it is sorted to the right one. This procedure terminates when a *leaf node* is reached. The final prediction of the ensemble $\boldsymbol{T}(x)$ is obtained by combining the predicted leaf values for each tree in the ensemble. In gradient boosting, the class probability is computed by applying a sigmoid transformation to the sum of the leaf values.

**Evasion Attacks**  An *evasion attack* involves manipulating valid inputs $x$ into adversarial examples $\tilde{x}$ in order to elicit a misprediction [23]. Following existing work on tree ensembles [23, 7, 10], we say that $\tilde{x}$ is an **adversarial example** for normal example $x$ when (1) $\|\tilde{x} - x\|_\infty < \delta$ where $\delta$ is a user-selected maximum distance (i.e., the two are sufficiently close), (2) the ensemble predicts the correct label for $x$, and (3) the model's predicted labels for $\tilde{x}$ and $x$ differ.

We briefly describe the two existing adversarial example generation methods $\mathcal{A} : (\boldsymbol{T}, x, \delta, t_{\max}) \to \{SAT(\tilde{x}), UNSAT, TIMEOUT\}$ used in this paper: *kantchelian* [23] and *veritas* [10]. These methods take as input an ensemble $\boldsymbol{T}$, a normal example $x$, a maximum perturbation size $\delta$, and a

timeout $t_{\max}$. They output $SAT(\tilde{x})$, where $\tilde{x}$ is an adversarial example for $x$, $UNSAT$, indicating that no adversarial example exists, or $TIMEOUT$, indicating that no result could be found within $t_{\max}$. Timeouts are explicitly handled because adversarial example generation is NP-hard [23].

*kantchelian* formulates the adversarial example generation task as a mixed-integer linear program (MILP) and uses a generic MILP solver (e.g., Gurobi [20]). Specifically, *kantchelian* directly minimizes the $\delta = \|x - \tilde{x}\|_\infty$ value. Given an example $x$, it computes:

$$\min_{\tilde{x}} \|x - \tilde{x}\|_\infty \quad \text{subject to} \quad \boldsymbol{T}(x) \neq \boldsymbol{T}(\tilde{x}). \tag{1}$$

This approach exploits the fact that a tree ensemble can be viewed as a set of linear (in)equalities. Three sets of MILP variables are used. *Predicate variables* $p_i$ represent the split conditions, i.e., each $p_i$ logically corresponds to a split on an attribute $f$: $p_i \equiv f < \tau$. *Leaf variables* $l_i$ indicate whether a leaf node is active. The *bound variable* $b$ represents the $l_\infty$ distance between the original example $x$ and the adversarial example $\tilde{x}$. Constraints between the variables encode the structure of the tree. A set of predicate consistency constraints encode the ordering between splits. For example, if two split values $\tau_1 < \tau_2$ appear in the tree for attribute $f$, and $p_1 \equiv f < \tau_1$ and $p_2 \equiv f < \tau_2$, then $p_1 \implies p_2$. Leaf consistency constraints enforce that a leaf is only active when the splits on the root-to-leaf path to that leaf are satisfied. Lastly, the mislabel constraint requires the output to be a certain class: for leaf values $v_i$, $\sum_i v_i l_i \lessgtr 0$. The objective directly minimizes the *bound variable*.

*veritas* improves upon *kantchelian* in terms of run time by formulating the adversarial example generation problem as a heuristic search problem in a graph representation of the ensemble (originally proposed by [8]). The nodes in this graph correspond to the leaves in the trees of the ensemble. Guided by a heuristic, the search then repeatedly selects compatible leaves. Leaves of two different trees are compatible when the conjunction of the split conditions along the root-to-leaf paths of the leaves are logically consistent. For a given $\delta$, *veritas* solves the following optimization problem:[1]

$$\underset{\tilde{x}}{\text{optimize}} \ \boldsymbol{T}(\tilde{x}) \quad \text{subject to} \quad \|x - \tilde{x}\|_\infty < \delta \tag{2}$$

The output of the model $\boldsymbol{T}(\tilde{x})$ is maximized when the target class for $\tilde{x}$ is positive, and minimized otherwise. While *veritas* can also directly optimize $\delta$, in this paper we will use a predefined $\delta$ for *veritas*. To the best of our knowledge, *veritas* is the fastest approximate evasion attack for tree ensembles (see Appendix B.1).

## 3  Method

Adversarial example generation methods are often applied in the following setting:

**Given**  a tree ensemble $\boldsymbol{T}$, a set of test examples $\mathcal{D}$, and a maximum perturbation size $\delta$
**Generate**  adversarial examples for each $x \in \mathcal{D}$.

The goal of this paper is to exploit the fact that adversarial examples are sequentially generated for each example in $\mathcal{D}$. By analyzing previously found adversarial examples, we aim to improve the efficiency of adversarial example generation algorithms by biasing the search towards the perturbations that are most likely to lead to an adversarial example.

Our hypothesis is that some parts of the ensemble are disproportionately sensitive to small perturbations, i.e., crossing the thresholds of split conditions in these parts of the ensemble results in large changes in the predicted value. Prior work has hypothesized that robustness is related to fragile features and that such features are included in models because learners search for any signal that improves predictive performance [21]. One would expect that the attributes used in the split conditions in these disproportionately sensitive parts are exploited by adversarial examples more frequently than other attributes. Figure 1 illustrates this point by showing how often each attribute is perturbed in a set of a 10 000 adversarial examples generated by *kantchelian* for two datasets. The bar plots distinguish among attributes are (1) never modified by any adversarial example (left), (2) modified by at least one but at most 5% of all adversarial examples (middle), and (3) modified by more than 5% of the adversarial examples (right). Less than 10% of the attributes are used by more than 5% of the adversarial examples. The figure shows that regularities exist in constructed adversarial examples: examples generated for different normal examples tend to exhibit perturbations to the same small set of attributes. Thus the two questions are how can one identify these frequently-modified attributes and how can algorithms exploit this knowledge to more quickly generate adversarial examples.

---

[1]We are abusing terminology: here, $\boldsymbol{T}(x)$ is the predicted probability. Previously, it was the predicted label.

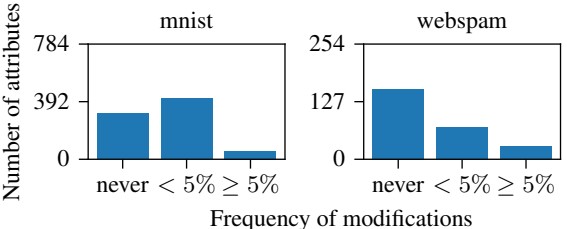

Figure 1: Bar plots showing that most attributes are not modified by the majority of adversarial examples on the *mnist* and *webspam* datasets. The leftmost bar shows the number of attributes that are never changed by any of the 10 000 adversarial examples generated by *kantchelian*'s approach. The middle bar shows the number of attributes that are modified at least once but at most by 5% of the adversarial examples. The rightmost bar shows the number of frequently modified features.

Our proposed approach has two parts. The first part simplifies the search for adversarial examples by only allowing perturbations to a limited subset of features. Namely, we exploit the knowledge that certain feature values are fixed to simplify the ensemble, by pruning away branches that can never be reached. The second part identifies a subset of commonly perturbed features by counting how often each feature is perturbed by adversarial examples. The size of this subset is determined by applying a theoretically grounded statistical test.

### 3.1 Modifying the Search Procedure

Our proposed approach speeds up the adversarial example generation procedure by limiting the scope of the adversarial perturbations to a subset of features $F_S$. This section assumes that we are given such a subset of features. The next section covers how to identify these features.

We consider three settings: *full*, *pruned*, and *mixed*. The *full* setting is the original configuration of *kantchelian* and *veritas*: the methods may perturb any attribute within a certain maximum distance $\delta$. That is, for each attribute $f \in F$ with value $x_f$, the attribute values are limited to $[x_f - \delta, x_f + \delta]$. Algorithm 1 summarizes the *pruned* and *mixed* approaches. We now describe both in greater detail.

---

**Algorithm 1** Fast repeated adversarial example generation

1: **parameters:** maximum perturbation size $\delta$, timeouts $t_{\max}^{full}$ and $t_{\max}^{prun}$ for *full* and *pruned*, genera-
   tion method $\mathcal{A} : (\boldsymbol{T}, x, \delta, t) \rightarrow \{SAT(\tilde{x}), UNSAT, TIMEOUT\}$
2: **function** GENERATE($\boldsymbol{T}_{full}, \mathcal{D}, F_S$, *mixed* flag)
3:     $\tilde{\mathcal{D}} \leftarrow \emptyset$
4:     **for** $x \in \mathcal{D}$ **do**
5:         $\boldsymbol{T}_{prun} \leftarrow$ PRUNE $(\boldsymbol{T}_{full}, F_S, x)$     (Sec. 3.1)
6:         $\alpha \leftarrow \mathcal{A}(\boldsymbol{T}_{prun}, x, \delta, t_{\max}^{prun})$
7:         **if** $\alpha \neq SAT(\tilde{x}) \wedge$ *mixed* flag set **then**
8:             $\alpha \leftarrow \mathcal{A}(\boldsymbol{T}_{full}, x, \delta, t_{\max}^{full})$
9:         **end if**
10:        $\tilde{\mathcal{D}} \leftarrow \tilde{\mathcal{D}} \cup \{\alpha\}$
11:     **end for**
12:     **return:** $\tilde{\mathcal{D}}$
13: **end function**

---

**Pruned Approach** The *pruned* setting disallows modifications to the attributes in the *non-selected* set of attributes $F_{NS} = F \setminus F_S$. We accomplish this by pruning the trees in the ensemble. Any node splitting on attributes in $F_{NS}$ is removed. Its parent node is directly connected to the only child node that can be reached by examples with the fixed value for the attribute. Figure 2 shows an example of this procedure. We refer to this procedure as PRUNE($\boldsymbol{T}, F_S, x$). The adversarial example generation methods can be applied as normal to the pruned ensemble, but they will only generate adversarial examples with perturbations to the attributes in $F_S$. Pruning simplifies the MILP problem of *kantchelian* because all predicate variables $p_i$ that correspond to splits in internal nodes of pruned

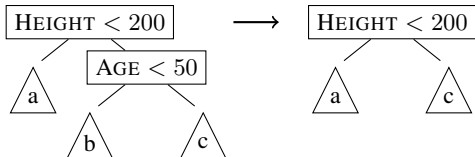

Figure 2: An example tree using two attributes HEIGHT and AGE (left). Suppose $F_{NS} = \{\text{AGE}\}$. Given an example where AGE $= 55$, we can prune away the internal node splitting on AGE. In the resulting tree (right), subtree (b) is pruned because it is unreachable given that AGE $= 55$ and only subtrees (a) and (c) remain.

subtrees, and leaf variables $l_i$ that correspond to leaves of pruned subtrees can be removed from the mathematical formulation. For *veritas*, the search space is reduced in size because the pruned leaves are removed from the graph representation of the ensemble. Hence, for both systems, on average, the problem difficulty is reduced by pruning the ensembles.

Pruning the trees does not affect the validity of generated adversarial examples: if $\tilde{x}$ is an adversarial example for a normal example $x$ generated on a pruned ensemble, then $\tilde{x}$ is also an adversarial example for the full ensemble.

**Proposition 3.1.** *Given normal example $x$ that is correctly classified by the full ensemble $\boldsymbol{T}_{full}$. Let $\boldsymbol{T}_{prun} = \text{PRUNE}(\boldsymbol{T}_{full}, F_S, x)$ and $\tilde{x} = \mathcal{A}(\boldsymbol{T}_{prun}, x, \delta, t_{\max})$ (i.e., $\boldsymbol{T}_{prun}(x) \neq \boldsymbol{T}_{prun}(\tilde{x})$ and $\|x - \tilde{x}\|_\infty < \delta$). Then it holds that $\boldsymbol{T}_{full}(x) \neq \boldsymbol{T}_{full}(\tilde{x})$.*

*Proof.* Because only branches not visited by $x$ are removed, $\boldsymbol{T}_{prun}(x) = \boldsymbol{T}_{full}(x)$. The values for features in $F_{NS}$ are fixed, so these values are equal between $x$ and $\tilde{x}$. Hence, $\tilde{x}$ only visits branches in $\boldsymbol{T}_{full}$ that are also in $\boldsymbol{T}_{prun}$. Therefore, $\boldsymbol{T}_{prun}(\tilde{x}) = \boldsymbol{T}_{full}(\tilde{x})$. $\qquad\square$

However, an $UNSAT$ generated on a pruned ensemble is inconclusive: it might still be the case that an adversarial example exists for the full ensemble, albeit one with perturbations to features in $F_{NS}$. The *pruned* setting generates a **false negative** if it reports $UNSAT$, yet the *full* setting reports $SAT$.

**Mixed Approach**  The *mixed* setting takes advantage of the fast adversarial generation capabilities of the *pruned* setting, but falls back to the *full* setting when the *pruned* setting returns an $UNSAT$ or times out. A much stricter timeout $t_{\max}^{prun}$ is used for the *pruned* setting to fully take advantage of the fast $SAT$s, while avoiding spending time on an uninformative $UNSAT$. The *mixed* setting is guaranteed to find an adversarial example if the *full* setting can find one.

**Theorem 3.2.** *Assume a normal example $x$ and maximum distance $\delta$. If an adversarial example can be found for the full ensemble $\boldsymbol{T}_{full}$, then the mixed setting is guaranteed to find an $\tilde{x}$ such that $\|x - \tilde{x}\|_\infty < \delta$ and $\boldsymbol{T}_{full}(x) \neq \boldsymbol{T}_{full}(\tilde{x})$.*

*Proof.* The *mixed* setting first operates on the pruned ensemble $\boldsymbol{T}_{prun}$ using a tight timeout and optimizes Equation 1 or 2 using *kantchelian* or *veritas* respectively. This returns (1) an adversarial example $\tilde{x}$, (2) an $UNSAT$ or (3) times out. In case (1), the generated adversarial example $\tilde{x}$ is also an adversarial example for the full ensemble (Prop 3.1). In cases (2) and (3), the *mixed* setting falls back to the *full* setting operating on the full ensemble $\boldsymbol{T}_{full}$ with the same timeout. Hence, it inherits the full method's guarantees. $\qquad\square$

## 3.2 Identifying Relevant Features

A good subset of relevant attributes $F_S$ should satisfy two properties. First, it should minimize the number of false negatives, which occur when the *pruned* approach reports $UNSAT$, but the *full* approach reports $SAT$. Second, the feature subset should be small. The smaller $F_S$ is, the more the ensemble can be pruned, and the larger the speedup is. These two objectives are somewhat in tension. Including more features will reduce the number of false negatives but limit the speedups, whereas using a very small subset will restrict the search too much resulting in many false negatives (or slow calls to the full search in the *mixed* setting). The procedure is given in Algorithm 2.

We address the first requirement by adding features to the subset that are frequently perturbed by adversarial examples. We rank features by counting how often each one differs between the perturbed adversarial examples in $\tilde{\mathcal{D}}$ so far and their corresponding normal examples in $\mathcal{D}$.

The second requirement is met by statistically testing whether the identified subset guarantees that the false negative rate is smaller than a given threshold $\tau$ with probability at least $1 - \eta$, for a specified confidence parameter $\eta$. If it is not guaranteed, then the subset is expanded. This is done at most 4 times for subsets of 5%, 10%, 20%, 30% of the features (EXPANDFEATURESET$(F_S, \mathcal{D}, \tilde{\mathcal{D}})$ in Algorithm 2). If all tests fail, then a final feature subset of 40% of the most commonly modified features is used. We do not go beyond 40% because using the full feature set is then more efficient. Each test is executed on a small set of $n$ generated adversarial examples. A first zeroth set is used merely for obtaining the first feature counts.

The statistical tests are performed as follows. The null hypothesis is that *FNR* is greater than the threshold $\tau$. Take $\mathcal{D}_F = (x_1, x_2, \ldots, x_N)$ the dataset we use to find the feature subset $F_S$. We define $\mathbf{v} = (v_1, v_2, \ldots, v_N)$ to be the binary vector such that $v_i = 1$ if the *pruned* search with the feature subset $F_S$ returns *UNSAT* for the example $x_i$ but the *full* search returns *SAT*, and $v_i = 0$ otherwise. Then the true false negative rate corresponding to $F_S$ can be written as $FNR = \frac{1}{N} \sum_{i=1}^{N} v_i$. The small set of $n$ examples from which we are estimating the false negative rate is a random vector $\mathbf{X} = (X_1, X_2, \ldots, X_n)$ sampled without replacement from $\mathcal{D}_F$. We also define $\mathbf{V} = (V_1, V_2, \ldots, V_n)$ where $V_i$ is the binary random variable defined analogically to how we defined $v_i$. It follows that $\sum_{i=1}^{n} V_i$ is distributed as a hypergeometric random variable. We use the method of *inversion of acceptance intervals* to find a one-sided confidence interval $[0; \Delta]$ for the false negative rate with confidence level equal to a given $1 - \eta$ (see, e.g., Section 5.2 in [2]), exploiting the fact that the cumulative distribution function of a hypergeometric distribution can be computed efficiently (CONFIDENCEINTERVAL$(\bar{v}, n, \eta)$ in Algorithm 2). We reject the null hypothesis if the confidence interval does not contain the threshold $\tau$. It follows from the basic properties of confidence intervals that this yields the desired test with confidence $1 - \eta$. Since we execute the test 4 times in the algorithm, we apply a union-bound correction of factor 4 (we use confidence level 0.9). Note that there is a trade-off. The higher $n$, the better the statistical estimates and the counts are, but also the more examples we process with a potentially suboptimal feature subset.

---

**Algorithm 2** Find feature subset

---

1: **parameters:** set of normal examples $\mathcal{D}$, sample size $n$, acceptable false negative rate $\tau$, confidence parameter $\eta$

2: $F_S \leftarrow \emptyset$
3: **for** $k \in 0..4$ **do**
4:      $\tilde{\mathcal{D}} \leftarrow$ GENERATE$(\boldsymbol{T}, \mathcal{D}[kn, k(n+1)], F_S, true)$
5:      $\bar{v} \leftarrow \frac{1}{n} \times$ number of false negatives in $\tilde{\mathcal{D}}$
6:      $[0; \Delta] \leftarrow$ CONFIDENCEINTERVAL$(\bar{v}, n, \eta)$
7:      **if** the threshold $\tau$ is in $[0; \Delta]$, **then** EXPANDFEATURESET$(F_S, \mathcal{D}, \tilde{\mathcal{D}})$
8:      **else break** the loop
9: **end for**
10: **return:** $F_S$

---

## 4 Experiments

Empirically, we address three questions: (Q1) Is our approach able to improve the run time of generating adversarial examples? (Q2) How does ensemble complexity affect our approach's performance? (Q3) What is our empirical false negative rate?

Because the described procedure is based on identifying a subset of relevant features, it makes sense to exploit it only when the dataset has a large number of dimensions. Therefore, we present numerical experiments for ten binary classification tasks on high-dimensional datasets, using both tabular data and image data, as shown in Table 1.

**Experimental Setup** We apply 5-fold cross validation for each dataset. We use four of the folds to train an XGBoost [9], random forest [3, 25] or GROOT forest (a robustified ensemble type [32])

ensemble $\boldsymbol{T}$. From the test set, we randomly sample 10 000 normal examples and attempt to generate adversarial examples by perturbing each one using the *kantchelian* or *veritas* attack. Table 1 also reports the adopted values of maximum perturbation $\delta$ and the hyperparameters of the learned ensembles, which were selected via tuning using the grid search described in Appendix B. The experiments were run on an Intel(R) E3-1225 CPU with 32GiB of memory.

Table 1: Datasets' characteristics: *N* and *#F* are the number of examples and the number of features. *higgs* and *prostate* are random subsets of the original, bigger datasets. Multi-class classification datasets were converted to binary classification: for *covtype* we predict majority-vs-rest, for *mnist* and *fmnist* we predict classes 0-4 vs. classes 5-9, and for *sensorless* classes 0-5 vs. classes 6-10. We also report adopted values of max allowed perturbation $\delta$ and learners' tuned hyperparameters after the grid search described in Appendix B. Each ensemble $\boldsymbol{T}$ has maximum tree depth d and contains M trees. The learning rate for XGBoost is $\eta$. GROOT robustness is defined by $\epsilon$.

| Dataset | N | #F | XGBoost | | | | RF | | | GROOT | | | |
| | | | $\delta$ | M | d | $\eta$ | $\delta$ | M | d | $\delta$ | M | d | $\epsilon$ |
|---|---|---|---|---|---|---|---|---|---|---|---|---|---|
| covtype | 581k | 54 | 0.1 | 50 | 6 | 0.9 | 0.3 | 50 | 10 | 0.4 | 50 | 10 | 0.01 |
| fmnist | 70k | 784 | 0.3 | 50 | 6 | 0.1 | 0.3 | 50 | 10 | 0.4 | 50 | 10 | 0.3 |
| higgs | 250k | 33 | 0.08 | 50 | 6 | 0.1 | 0.08 | 50 | 10 | 0.4 | 50 | 10 | 0.01 |
| miniboone | 130k | 51 | 0.08 | 50 | 6 | 0.1 | 0.08 | 50 | 10 | 0.5 | 50 | 10 | 0.01 |
| mnist | 70k | 784 | 0.3 | 50 | 6 | 0.5 | 0.3 | 50 | 10 | 0.4 | 50 | 10 | 0.3 |
| prostate | 100k | 103 | 0.1 | 50 | 4 | 0.5 | 0.2 | 50 | 10 | 0.2 | 50 | 10 | 0.01 |
| roadsafety | 111k | 33 | 0.06 | 50 | 6 | 0.5 | 0.12 | 50 | 10 | 0.2 | 50 | 10 | 0.05 |
| sensorless | 58.5k | 48 | 0.06 | 50 | 6 | 0.5 | 0.12 | 50 | 10 | 0.2 | 50 | 10 | 0.01 |
| vehicle | 98k | 101 | 0.15 | 50 | 6 | 0.1 | 0.15 | 50 | 10 | 0.4 | 50 | 10 | 0.1 |
| webspam | 350k | 254 | 0.04 | 50 | 5 | 0.5 | 0.06 | 50 | 10 | 0.1 | 50 | 10 | 0.01 |

The *pruned* and *mixed* settings work as follows. We use the procedure from Section 3.2 to select a subset of relevant features. We use $\tau = 0.25$, $n = 100$ and $\eta = 0.1$. We then apply Algorithm 2: we generate 5 sets of $n$ adversarial examples to (1) find which features are perturbed most often and (2) determine the size of the feature subset $F_S$. Therefore the extracted feature set gives us a $1-\eta = 90\%$ confidence that our true false negative rate is below $25\%$. After Algorithm 2 terminates, $F_S$ is fixed, and we run the *pruned* and *mixed* settings on all the remaining test examples (Algorithm 1). We set a timeout of one minute for the *full* setting, and a much stricter timeout of 1 (*kantchelian*) or 0.1 (*veritas*) seconds in the *pruned* setting. We can be stricter with *veritas* as it is an approximate method that is faster than the exact *kantchelian*.[2]

**Q1: Run Time** Table 2 reports the average run time for the *full* setting and the average speedup given by the *pruned* and *mixed* settings. We present here results for XGBoost and random forest, and report results for GROOT together with more extended results in Appendix C. Considering all three model types and both attacks, speedups for the *pruned* setting are in the range 1.4x-36.2x with an average of 9x ($\pm$ 8x), and for the *mixed* in the range 1.1x-20.5x with an average of 4x ($\pm$ 3x).

We notice that generating adversarial examples is more difficult for random forests (RF) than XGB. This leads to our strategies offering larger wins for RF than for XGB, with average speedups of 9.4x/3.5x ($\pm$ 7.2x/1.6x) for RF and 4.7x/2.7x ($\pm$ 3.4x/1.2x) for XGB. The robustified GROOT forests are even harder to attack, meaning our methods offer even larger improvements with average speedups of 11.4x/4.9x ($\pm$ 9.9x/5.0x).[3]

Tables 5 and 6 in the supplement also report additional statistics on the presented experiments. On average, the *mixed* setting falls back to the *full* search $10.5\%$ of the time. The model and attack type do not seem to have a strong influence on the proportion of calls to the *full* search. This helps it achieve a speedup by taking advantage of the fast $SAT$ results of the *pruned* setting while still offering the theoretical guarantee from Theorem 3.2.

We also report the attack success rate, which is the fraction of times where our methods generate an adversarial example given that a valid adversarial example exists for the full model. The *pruned* search has an average success rate of $90\%$ ($\pm$ $6\%$). The *mixed* search has success rate $100\%$ by definition (Theorem 3.2).

---

[2]Appendix D provides a sensitivity analysis for the hyperparameter settings of the statistical test and timeouts.
[3]See Tables 5 and 6 in the supplement.

Table 2: Average run times and speedups when attempting to generate 10 000 adversarial examples using *kantchelian/veritas* on an XGBoost/random forest ensemble for *full*, *pruned* and *mixed*. A * means that the dataset exceeded the global timeout of six hours.

| | Kantchelian XGB | | | Kantchelian RF | | | Veritas XGB | | | Veritas RF | | |
|---|---|---|---|---|---|---|---|---|---|---|---|---|
| | *full* | *pruned* | *mixed* | *full* | *pruned* | *mixed* | *full* | *pruned* | *mixed* | *full* | *pruned* | *mixed* |
| covtype | 9.9m | 3.3× | 2.1× | 25.7m | 11.0× | 3.7× | 5.3s | 1.7× | 1.4× | 45.0s | 6.0× | 2.9× |
| fmnist | 1.5h | 4.9× | 3.9× | 56.2m | 7.8× | 4.3× | 43.6s | 1.4× | 1.3× | 6.6m | 3.6× | 3.0× |
| higgs | 3.3h | 4.1× | 1.8× | 5.1h | 3.0× | 1.4× | 20.4s | 2.9× | 2.4× | 41.8m | 21.4× | 2.5× |
| miniboone | 6.0h* | 10.9× | 3.4× | 6.0h* | 9.4× | 5.2× | 1.2m | 8.4× | 5.9× | 12.9m | 11.8× | 6.4× |
| mnist | 23.9m | 6.9× | 5.1× | 36.6m | 5.7× | 4.8× | 47.9s | 2.5× | 2.1× | 3.3m | 3.2× | 2.9× |
| prostate | 12.8m | 3.4× | 2.8× | 6.0h* | 11.8× | 5.2× | 9.9s | 2.5× | 2.2× | 23.7m | 16.9× | 2.6× |
| roadsafety | 10.7m | 3.0× | 2.0× | 45.2m | 5.4× | 2.3× | 11.4s | 2.7× | 2.1× | 40.6m | 33.5× | 3.1× |
| sensorless | 29.8m | 2.3× | 2.1× | 52.5m | 5.7× | 3.6× | 12.1s | 2.9× | 1.8× | 4.1m | 4.7× | 1.5× |
| vehicle | 2.5h | 5.9× | 3.4× | 3.8h | 7.1× | 5.8× | 19.4m | 15.6× | 1.9× | 42.8m | 9.1× | 1.1× |
| webspam | 24.2m | 5.7× | 3.7× | 1.5h | 7.3× | 5.7× | 18.8s | 2.6× | 2.1× | 12.9m | 3.9× | 1.2× |

Figure 3 shows the number of executed searches as a function of time for four combinations of attack algorithm and model type.[4] For XGB, both attacks benefit. Moreover, the *mixed* setting is typically very close in run time to the *pruned*. On RF, the *pruned* setting offers larger speedups. However, we see a more noticeable difference between the *pruned* and the *mixed* search on several datasets. This indicates that the *mixed* strategy falls back more often to an expensive *full* search.

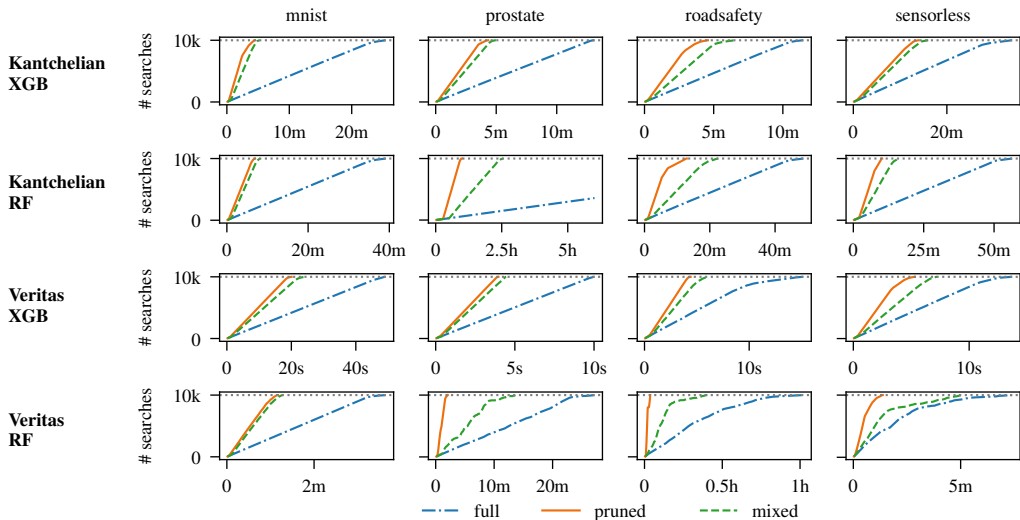

Figure 3: Average run times for 10 000 calls to *full*, *pruned* and *mixed* for *kantchelian* (top) and *veritas* (bottom). Results are given for both XGBoost and random forest for four selected datasets.

Finally, it is natural to wonder how the quality of the generated adversarial examples is affected by the modified search procedure. While this is difficult to quantify, Figure 4 provides some examples of constructed adversarial examples for the *mnist* dataset and an XGBoost ensemble. Visually, the examples constructed by *full* and *pruned* settings for both attacks are very similar. The examples constructed using *kantchelian* look more similar to the base example than those for *veritas* because *kantchelian* finds the closest possible adversarial example whereas *veritas* has a different objective: it constructs an adversarial example that will elicit a highly confident misprediction. See Appendix E for more generated examples.

**Q2: Scaling Behavior** Two key hyperparameters of tree ensembles are the maximum depth of each learned tree and the number of trees in the ensemble. We explore how varying these affects our approach, employing the same setup as described in Q1. We use the *mnist* dataset and omit

---

[4]The supplement shows these plots for all datasets plus for GROOT forests (see Appendix C).

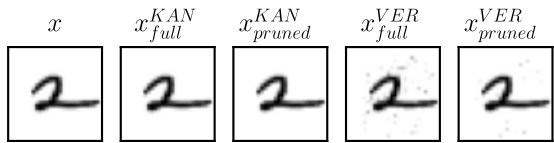

Figure 4: Generated adversarial examples for an *mnist* digit and an XGBoost ensemble, using both attacks (*full* vs *pruned*).

*kantchelian* with RFs due to its computational cost. Figure 5 (top) shows how the run time to perform 10 000 searches varies as function of the maximum tree depth for a fixed ensemble size of 50. The run times for the *pruned* and *mixed* approaches grow very slowly as the depths are increased. In contrast, the *full* search scales worse: deeper trees lead to higher run times. Figure 5 (bottom) shows how the run time to perform 10 000 searches varies as function of the ensemble size for a fixed maximum tree depth of 6 for XGB and 10 for RF. Again, the *pruned* and *mixed* approaches show much better scaling behavior. Note that *veritas*'s full search shows a very large jump on RF when moving from 75 to 100 trees. These results indicate that our approaches will offer even better run time performance than the standard full search for more complex ensembles.

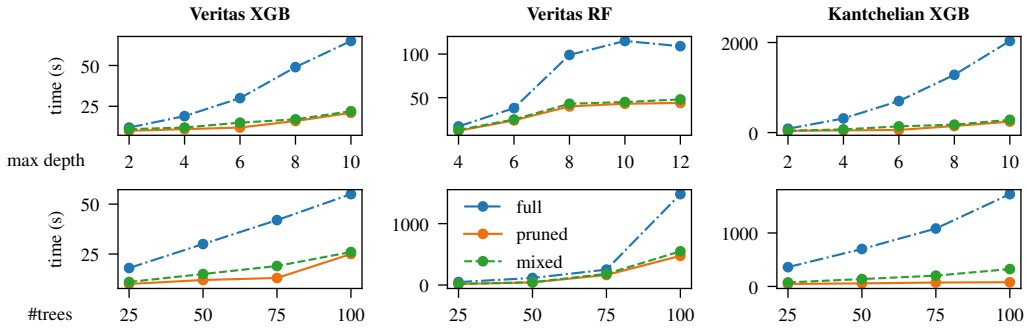

Figure 5: Run time of *full*, *mixed* and *pruned* settings using *veritas* XGB, *veritas* RF, *kantchelian* XGB on *mnist*, and varying the max depth (top) and number of estimators in the ensemble (bottom).

**Q3: Empirical FNR**  We use Algorithm 2 to bound the false negative rate to be less than 25% with high probability. Tables 5 and 6 in the supplement report the empirical false negative rates for all experiments. The average false negative rate is 7.5% and the maximum is 17.1%. Hence, empirically we achieve better results than the theory guarantees. Neither the ensemble method nor the attack type strongly influence the false negative rate. These small false negative rates still allow dramatically reducing the number of considered features. On average, $F_S$ contains 17% of the features. Out of 300 experiments,[5] we only select the maximum percentage of features 17 times. Generally, *kantchelian* requires slightly more features than *veritas* and RF models requires slightly more features than XGB/GROOT models.

To provide a better intuition on the relationship between the empirical FNR and the speedup, Figure 6 shows this tradeoff on two datasets using XGBoost. In essence, higher FNRs correspond to smaller feature subsets, hence larger speedups.

## 5   Related Work

Adversarial examples have been theoretically studied and defined in multiple different ways [14, 18].

Approaches to reason about learned tree ensembles have received substantial interest in recent years including algorithms to perform evasion attacks [23, 15] (i.e., generate adversarial examples),

---

[5] 5 folds x 10 datasets x 3 ensemble types x 2 attacks

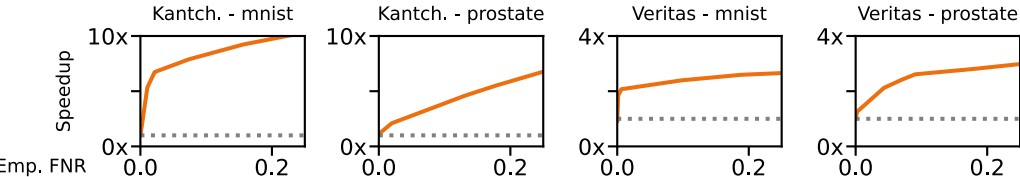

Figure 6: Speedups achieved by the *pruned* setting when attempting to generate 10 000 adversarial examples using *kantchelian* (left) and *veritas* (right) on an XGB ensemble, varying the empirical false negative rate. The dotted horizontal line corresponds to a speedup of 1x, i.e., same run time of the *full* setting.

perform robustness checking [8], and verify that the ensembles satisfy certain criteria [11, 10, 26, 29]. Kantchelian et al. [23] were the first to show that tree ensembles are susceptible to evasion attacks. Their MILP formulation is still the most frequently used method to check robustness and generate adversarial examples. Beyond this exact approach, several approximate approaches exist [8, 10, 35, 36] though not all of them are able to generate concrete adversarial examples (e.g., [8, 35]).

Other work focuses on making tree ensembles more robust. Approaches for this include adding generated adversarial examples to the training data (model hardening) [23], or modifying the splitting procedure [7, 4, 32]. Gaining further insights into how evasion attacks target tree ensembles, like those contained in this paper, may inspire novel ways to improve the robustness of learners.

Another line of work aims at directly training tree ensembles that admit verification in polynomial time [5, 13]. However, a drawback to current approaches is that they result in (large) decreases in predictive performance.

Finally, performing evasion attacks has been studied for other model classes with deep neural networks receiving particular attention [28, 17, 24, 6]. However, state-of-the-art algorithms are tailored to one specific model type as they typically exploit specific properties of the model, e.g., the work on tree ensembles often exploits the logical structure of a decision tree.

# 6   Conclusions

This paper explored two methods to efficiently generate adversarial examples for tree ensembles. We showed that considering only the *same subset of features* is typically sufficient to generate adversarial examples for tree ensemble models. We proposed a simple procedure to quickly identify such a subset of features, and two generic approaches that exploit it to speed up adversarial examples generation. We showed how to apply them to an exact (*kantchelian*) and approximate (*veritas*) evasion attack on three types of tree ensembles, and discussed their properties and run time performances.

**Limitations.** Our approach speeds up evasion attacks in the specific scenario when the same model is repeatedly attacked. Plus, it excels on high-dimensional datasets. Our evaluation only considered $l_\infty$ attacks, whereas other norms such $l_1$ and $l_2$ are also relevant.

**Impact Statement.** While this work does make attacking tree ensembles faster, it is also important to understand what attackers may do. This work also targets increasing the applicability of robustness checking and hardening techniques, which can lead to approaches for training more robust models.

**Acknowledgements.** This research is supported by the Research Foundation Flanders (FWO, LC: 11I8125N), The European Union's Horizon Europe Research and Innovation program under the grant agreement TUPLES No. 101070149 (LC, LD, OK, JD), and the Flemish Government under the "Onderzoeksprogramma Artificële Intelligentie (AI) Vlaanderen" program (JD).

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

# A   Analysis of the Problem Setting

Adversarial examples are often generated for tasks like computing adversarial accuracy [32], computing empirical robustness [10], and performing model hardening [23]. The effect of using the approximation proposed in this paper differs for each task.

Computing the adversarial accuracy of a classifier only requires determining whether an adversarial example $\tilde{x}$ exists within the given $\delta$ for each provided normal example $x$. Because the *mixed* strategy reverts to the original complete search when the *pruned* approach returns an $UNSAT$, as stated in Theorem 3.2 it is guaranteed to find an adversarial example if it exists. Hence, the *mixed* strategy can speed up computing the adversarial accuracy without affecting its value.

Computing the empirical robustness of a classifier requires finding the nearest adversarial example $\tilde{x}$ for each normal example $x$. Because the *pruned* approach does not consider all features and the *mixed* approach may not, they may return an adversarial example that is further away than if the full search space was considered. Hence, when using an exact attack like *kantchelian*, the empirical robustness computed using the *mixed* strategy is an overestimate of the true empirical robustness. We show this and we study what happens with an approximate method in Appendix E.

In model hardening, a large number of adversarial examples are generated and added to the training data [23]. The *pruned* approach can be used to generate a lot more adversarial examples in a fixed amount of time.

# B   Employed Datasets, Models and Attacks

We expand the discussion on Table 1, which reports the characteristics of the employed datasets and models. In our experiments, we use ten high-dimensional datasets where the number of dimensions is greater than 25. We verified that our approach does not bring consistent run time improvements for datasets with less than 25 features, or with a fully categorical domain (where the $l_\infty$ norm loses meaning).

Table 1 also reports the hyperparameters of the learned ensembles, which are tuned through grid search. For all model types, we choose the number of trees in $\{10, 20, 50\}$. Max depth is chosen in the range $[3, 6]$ for XGBoost, and in $\{5, 7, 10\}$ for random forest and GROOT forest (which typically need deeper trees to work better). XGBoost learning rate is chosen among $\{0.1, 0.5, 0.9\}$. GROOT forest $\epsilon$ is chosen among $\{0.01, 0.05, 0.1, 0.3, 0.5\}$, and we select the model with the largest $\epsilon$ such that GROOT forest accuracy does not drop below 90% of the corresponding RF accuracy. Note that in GROOT a bigger $\epsilon$ corresponds to a more robust model, hence accuracy drops up to a point where the model can become useless in practice.

When running *kantchelian* on random forests and GROOT forests, we had to limit the number of estimators to 25 due to the extremely long run times.

While the model sizes are smaller, these ensembles are already challenging for the full settings of *kantchelian* and *veritas*. This is also highlighted in Q2 from Section 4 where we empirically study the effect of increasing the ensemble size on performance. Those results show that the *full* procedures becomes increasingly slower as the ensemble complexity grows, and our method offers larger wins.

Table 3 gives specific reference to each of the considered datasets.

## B.1   Comparison with LT Attack

We evaluate our two methods on a representative set of scenarios, varying the model type (XGBoost, random forest, GROOT forest), and the evasion attack variant (one exact (*kantchelian*) and one approximate (*veritas*)). Given the fact that (1) it is reasonable to expect that most evasion attacks will benefit from smaller pruned models, and (2) we see improvement across all these settings, we are confident that run time improvements also translate to other evasion attack methods.

In practice, other approximate attacks alternative to *veritas* exist. To the best of our knowledge, no alternative outperforms *veritas* run times. Computationally, we have compared *veritas* to another popular state-of-the-art method: LT-attack [36]. On the full setting, they have the same success rate and *veritas* is 25 to 60 times faster, as shown in Table 4 for XGBoost ensembles.

Table 3: References to all the ten datasets used in the experiments.

| Dataset | link |
|---------|------|
| covtype | `https://www.openml.org/d/1596` |
| fmnist | `https://www.openml.org/d/40996` |
| higgs | `https://www.openml.org/d/42769` |
| miniboone | `https://www.openml.org/d/44128` |
| mnist | `https://www.openml.org/d/554` |
| prostate | `https://www.openml.org/d/45672` |
| roadsafety | `https://www.openml.org/d/45038` |
| sensorless | `https://archive.ics.uci.edu/dataset/325` |
| vehicle | `https://www.openml.org/d/357` |
| webspam | `https://www.csie.ntu.edu.tw/~cjlin/libsvmtools/datasets/binary.html#webspam` |

Table 4: Average run times for generating $10\,000$ adversarial examples using *LT-attack* and *veritas* in the *full* search setting.

|  | *mnist* | *prostate* | *roadsafety* | *sensorless* |
|---|---|---|---|---|
| *LT* | 48m | 10.5m | 5.0m | 7.1m |
| *veritas* | 0.8m | 0.2m | 0.2m | 0.2m |

## C  Expanded Experimental Results

Tables 5 (*kantchelian*) and 6 (*veritas*) report the average times and speedups when attempting to generate $10\,000$ adversarial examples in each of our experimental scenarios. The averages are computed over five folds. There is one table for each combination of attack (*kantchelian*, *veritas*) and ensemble type (XGB, random forest, GROOT forest). For each dataset, we also report the average size of the relevant feature subset $F_S$, the percent of searches in the *mixed* setting that require making a call to the *full* search, the false negative rate (proportion of times that *pruned* returns $UNSAT$ but *full* returns $SAT$), the *pruned* setting success rate (i.e., an adversarial example can be generated with the *full* search, and the *pruned* setting finds a valid adversarial example), and the percent of examples that were skipped due to the *full* search reaching the global timeout of six hours. The *pruned* and *mixed* settings never reach the global timeout. Note that the *mixed* setting has attack success rate of 100% by definition.

Figures 7 (*kantchelian*) and 8 (*veritas*) show the number of executed searches as a function of time for *kantchelian* and *veritas* on all ten datasets. Each plot contains the results for XGB (top), RF (middle) and GROOT forest (bottom). Hence these plots show the complete set of results from Figure 3 in the main paper.

### C.1  Run Time Standard Deviation and Timeouts

Tables 7 (*kantchelian*) and 8 (*veritas*) extend the run time results of the presented experiments by additionally reporting standard deviations.

Tables 7 (*kantchelian*) and 8 (*veritas*) also show the percentage of searches that timed out for each dataset, ensemble type and method. In short, XGBoost ensembles are on average easier to verify, and the searches almost never time out. On the other hand, random forests and GROOT forests are more challenging. It can happen that with a strict timeout, the *pruned* setting is not able to find a solution, as the task remains complex even working with a reduced feature set. In those cases, *pruned* ends with a $TIMEOUT$ and *mixed* will have to execute the full search.

### C.2  Tradeoff FNR vs Speedup

We further extend Figure 6 from Q3 in Section 4, which explicitly shows the relationship between the empirical false negative rate and the speedup of the *pruned* setting.

Table 5: Average run times (and speedups) when attempting to generate 10 000 adversarial examples using *kantchelian* on an XGBoost/random forest/GROOT forest ensemble for all three approaches: *full*, *pruned* and *mixed*. We also report the average size of the relevant feature subset, the number of calls to the *full* setting during *mixed* (= number of *UNSAT* + number of *TIMEOUT* for *pruned*), the number of false negatives (*pruned* returns *UNSAT*, but *full* returns *SAT*), the *pruned* attack success rate (i.e., *pruned* succeeds in generating an adversarial example if an example can be generated for the full ensemble), and the percent of examples that were skipped due to the *full* search reaching the global timeout of six hours. Experiments that exceeded the timeout are starred. The *pruned* and *mixed* settings never reach the global timeout.

**Kantchelian, XGBoost**

| | *full* | *pruned* | | *mixed* | | % rel. feats | % full calls | % false neg. | % *pruned* attack success rate | % *full skip* |
|---|---|---|---|---|---|---|---|---|---|---|
| covtype | 9.9m | 3.0m | 3.3× | 4.7m | 2.1× | 5.6% | 11.7% | 11.3% | 88.7% | 0 % |
| fmnist | 1.5h | 18.5m | 4.9× | 22.9m | 3.9× | 14.6% | 3.9% | 3.9% | 96.1% | 0 % |
| higgs | 3.3h | 47.7m | 4.1× | 1.8h | 1.8× | 18.0% | 20.0% | 9.8% | 80.4% | 0 % |
| miniboone | 6.0h* | 55.5m | 9.0× | 2.8h | 3.4× | 33.2% | 9.2% | 4.3% | 91.7% | 43.1% |
| mnist | 23.9m | 3.4m | 6.9× | 4.7m | 5.1× | 8.8% | 4.0% | 4.0% | 96.0% | 0 % |
| prostate | 12.8m | 3.8m | 3.4× | 4.6m | 2.8× | 11.8% | 7.6% | 6.6% | 93.3% | 0 % |
| roadsafety | 10.7m | 3.6m | 3.0× | 5.4m | 2.0× | 23.1% | 12.9% | 12.7% | 87.2% | 0 % |
| sensorless | 29.8m | 12.8m | 2.3× | 14.4m | 2.1× | 33.8% | 7.4% | 7.0% | 92.9% | 0 % |
| vehicle | 2.5h | 25.7m | 5.9× | 44.2m | 3.4× | 27.4% | 12.2% | 12.2% | 87.8% | 0 % |
| webspam | 24.2m | 4.3m | 5.7× | 6.6m | 3.7× | 7.2% | 8.7% | 8.7% | 91.3% | 0 % |

**Kantchelian, RF**

| | *full* | *pruned* | | *mixed* | | % rel. feats | % full calls | % false neg. | % *pruned* attack success rate | % *full skip* |
|---|---|---|---|---|---|---|---|---|---|---|
| covtype | 25.7m | 2.3m | 11.0× | 6.9m | 3.7× | 5.6% | 11.0% | 8.5% | 91.3% | 0 % |
| fmnist | 56.2m | 7.2m | 7.8× | 13.0m | 4.3× | 14.0% | 8.8% | 8.8% | 91.2% | 0 % |
| higgs | 5.1h | 1.7h | 3.0× | 3.6h | 1.4× | 30.0% | 27.6% | 7.1% | 73.0% | 0 % |
| miniboone | 6.0h* | 52.0m | 9.4× | 1.9h | 5.2× | 32.0% | 5.3% | 2.1% | 95.3% | 56.1% |
| mnist | 36.6m | 6.4m | 5.7× | 7.6m | 4.8× | 10.5% | 2.4% | 2.4% | 97.6% | 0 % |
| prostate | 6.0h* | 56.4m | 11.8× | 2.5h | 5.2× | 11.4% | 13.9% | 4.2% | 88.3% | 64.2% |
| roadsafety | 45.2m | 8.4m | 5.4× | 19.7m | 2.3× | 27.1% | 14.8% | 13.5% | 86.3% | 0 % |
| sensorless | 52.5m | 9.2m | 5.7× | 14.8m | 3.6× | 32.6% | 10.9% | 10.9% | 89.1% | 0 % |
| vehicle | 3.8h | 31.8m | 7.1× | 39.3m | 5.8× | 43.0% | 2.6% | 2.6% | 97.4% | 0 % |
| webspam | 1.5h | 12.4m | 7.3× | 15.9m | 5.7× | 10.2% | 2.6% | 2.6% | 97.4% | 0 % |

**Kantchelian, GROOT**

| | *full* | *pruned* | | *mixed* | | % rel. feats | % full calls | % false neg. | % *pruned* attack success rate | % *full skip* |
|---|---|---|---|---|---|---|---|---|---|---|
| covtype | 8.1m | 1.9m | 4.3× | 2.4m | 3.4× | 6.2% | 3.8% | 3.5% | 96.5% | 0 % |
| fmnist | 3.1h | 15.6m | 11.7× | 21.4m | 8.6× | 5.3% | 2.7% | 2.7% | 97.3% | 0 % |
| higgs | 6.0h* | 2.0h | 3.6× | 4.6h | 1.6× | 41.1% | 25.6% | 5.8% | 74.5% | 16.8% |
| miniboone | 2.0h | 17.8m | 6.8× | 20.9m | 5.8× | 20.7% | <1 % | <1 % | 99.6% | 0 % |
| mnist | 50.0m | 5.5m | 9.2× | 16.3m | 3.1× | 5.4% | 13.3% | 13.3% | 86.7% | 0 % |
| prostate | 6.0h* | 1.2h | 10.3× | 2.8h | 4.5× | 11.4% | 14.7% | 5.0% | 86.5% | 61.7% |
| roadsafety | 6.7m | 1.5m | 4.4× | 4.2m | 1.6× | 35.4% | 34.3% | 15.2% | 81.3% | 0 % |
| sensorless | 55.6m | 8.6m | 6.4× | 12.7m | 4.4× | 13.2% | 10.9% | 10.6% | 89.4% | 0 % |
| vehicle | 2.7h | 29.7m | 5.5× | 53.8m | 3.0× | 23.3% | 9.0% | 8.9% | 91.0% | 0 % |
| webspam | 1.8h | 18.6m | 5.8× | 24.1m | 4.5× | 10.9% | 7.1% | 7.1% | 92.9% | 0 % |

Table 6: Average run times (and speedups) when attempting to generate 10 000 adversarial examples using *veritas* on an XGBoost/random forest/GROOT forest ensemble for all three approaches: *full*, *pruned* and *mixed*. We also report the average size of the relevant feature subset, the number of calls to the *full* setting during *mixed* (= number of $UNSAT$ + number of $TIMEOUT$ for *pruned*), the number of false negatives (*pruned* returns $UNSAT$, but *full* returns $SAT$), the *pruned* attack success rate (i.e., *pruned* succeeds in generating an adversarial example if an example can be generated for the full ensemble), and the percent of examples that were skipped due to the *full* search reaching the global timeout of six hours. Experiments that exceeded the timeout are starred. The *pruned* and *mixed* settings never reach the global timeout.

**Veritas, XGBoost**

|  | *full* | *pruned* |  | *mixed* |  | % rel. feats | % full calls | % false neg. | % *pruned* attack success rate | % *full skip* |  |
|---|---|---|---|---|---|---|---|---|---|---|---|
| covtype | 5.3s | 3.1s | 1.7× | 3.8s | 1.4× | 5.9% | 12.1% | 11.7% | 88.3% | 0 | % |
| fmnist | 43.6s | 31.9s | 1.4× | 33.6s | 1.3× | 9.3% | 3.8% | 3.7% | 96.2% | 0 | % |
| higgs | 20.4s | 7.1s | 2.9× | 8.6s | 2.4× | 20.0% | 3.4% | 2.9% | 97.1% | 0 | % |
| miniboone | 1.2m | 8.5s | 8.4× | 12.0s | 5.9× | 14.8% | 4.0% | 3.8% | 96.2% | 0 | % |
| mnist | 47.9s | 19.0s | 2.5× | 22.4s | 2.1× | 5.2% | 7.5% | 7.5% | 92.5% | 0 | % |
| prostate | 9.9s | 3.9s | 2.5× | 4.4s | 2.2× | 11.2% | 7.1% | 6.1% | 93.8% | 0 | % |
| roadsafety | 11.4s | 4.2s | 2.7× | 5.4s | 2.1× | 31.9% | 9.8% | 9.7% | 90.3% | 0 | % |
| sensorless | 12.1s | 4.2s | 2.9× | 6.6s | 1.8× | 17.5% | 15.1% | 14.7% | 85.2% | 0 | % |
| vehicle | 19.4m | 1.2m | 15.6× | 10.0m | 1.9× | 18.6% | 13.9% | 13.6% | 86.2% | 0 | % |
| webspam | 18.8s | 7.3s | 2.6× | 9.0s | 2.1× | 5.3% | 10.2% | 10.2% | 89.8% | 0 | % |

**Veritas, RF**

|  | *full* | *pruned* |  | *mixed* |  | % rel. feats | % full calls | % false neg. | % *pruned* attack success rate | % *full skip* |  |
|---|---|---|---|---|---|---|---|---|---|---|---|
| covtype | 45.0s | 7.5s | 6.0× | 15.4s | 2.9× | 5.9% | 5.7% | 4.4% | 95.5% | 0 | % |
| fmnist | 6.6m | 1.8m | 3.6× | 2.2m | 3.0× | 10.7% | 10.9% | 7.4% | 89.1% | 0 | % |
| higgs | 41.8m | 2.0m | 21.4× | 16.8m | 2.5× | 31.3% | 8.8% | 6.8% | 92.0% | 0 | % |
| miniboone | 12.9m | 1.1m | 11.8× | 2.0m | 6.4× | 19.2% | 6.0% | 4.8% | 94.2% | 0 | % |
| mnist | 3.3m | 1.1m | 3.2× | 1.2m | 2.9× | 10.5% | 4.8% | 3.5% | 95.2% | 0 | % |
| prostate | 23.7m | 1.4m | 16.9× | 9.3m | 2.6× | 13.1% | 11.5% | 10.0% | 89.1% | 0 | % |
| roadsafety | 40.6m | 1.2m | 33.5× | 13.0m | 3.1× | 18.8% | 11.6% | 11.1% | 88.9% | 0 | % |
| sensorless | 4.1m | 52.3s | 4.7× | 2.8m | 1.5× | 21.2% | 12.1% | 11.2% | 87.9% | 0 | % |
| vehicle | 42.8m | 4.7m | 9.1× | 38.9m | 1.1× | 23.0% | 32.8% | 17.1% | 67.3% | 0 | % |
| webspam | 12.9m | 3.3m | 3.9× | 10.6m | 1.2× | 10.6% | 13.9% | 2.8% | 86.2% | 0 | % |

**Veritas, GROOT**

|  | *full* | *pruned* |  | *mixed* |  | % rel. feats | % full calls | % false neg. | % *pruned* attack success rate | % *full skip* |  |
|---|---|---|---|---|---|---|---|---|---|---|---|
| covtype | 15.9s | 5.8s | 2.8× | 7.7s | 2.1× | 5.6% | 4.7% | 4.4% | 95.6% | 0 | % |
| fmnist | 32.2m | 53.4s | 36.2× | 1.6m | 20.5× | 5.5% | 2.1% | 1.6% | 97.9% | 0 | % |
| higgs | 4.1h* | 11.6m | 21.8× | 31.5m | 8.1× | 24.7% | 7.9% | 3.8% | 90.8% | 16.4% |  |
| miniboone | 2.6m | 6.1s | 25.8× | 9.7s | 16.2× | 7.2% | 2.6% | 2.6% | 97.4% | 0 | % |
| mnist | 3.1m | 1.8m | 1.7× | 2.5m | 1.2× | 6.2% | 11.8% | 7.7% | 88.2% | 0 | % |
| prostate | 33.9m | 4.2m | 8.1× | 15.7m | 2.2× | 12.5% | 19.7% | 7.3% | 80.4% | 0 | % |
| roadsafety | 1.0m | 11.4s | 5.5× | 48.6s | 1.3× | 39.4% | 18.7% | 16.4% | 83.2% | 0 | % |
| sensorless | 3.4m | 8.6s | 23.8× | 1.9m | 1.8× | 10.0% | 11.5% | 11.1% | 88.8% | 0 | % |
| vehicle | 4.0h* | 8.2m | 29.6× | 2.1h | 2.0× | 19.0% | 20.9% | 9.7% | 79.4% | 1.4% |  |
| webspam | 9.5m | 2.2m | 4.3× | 4.6m | 2.1× | 9.0% | 14.8% | 8.2% | 85.2% | 0 | % |

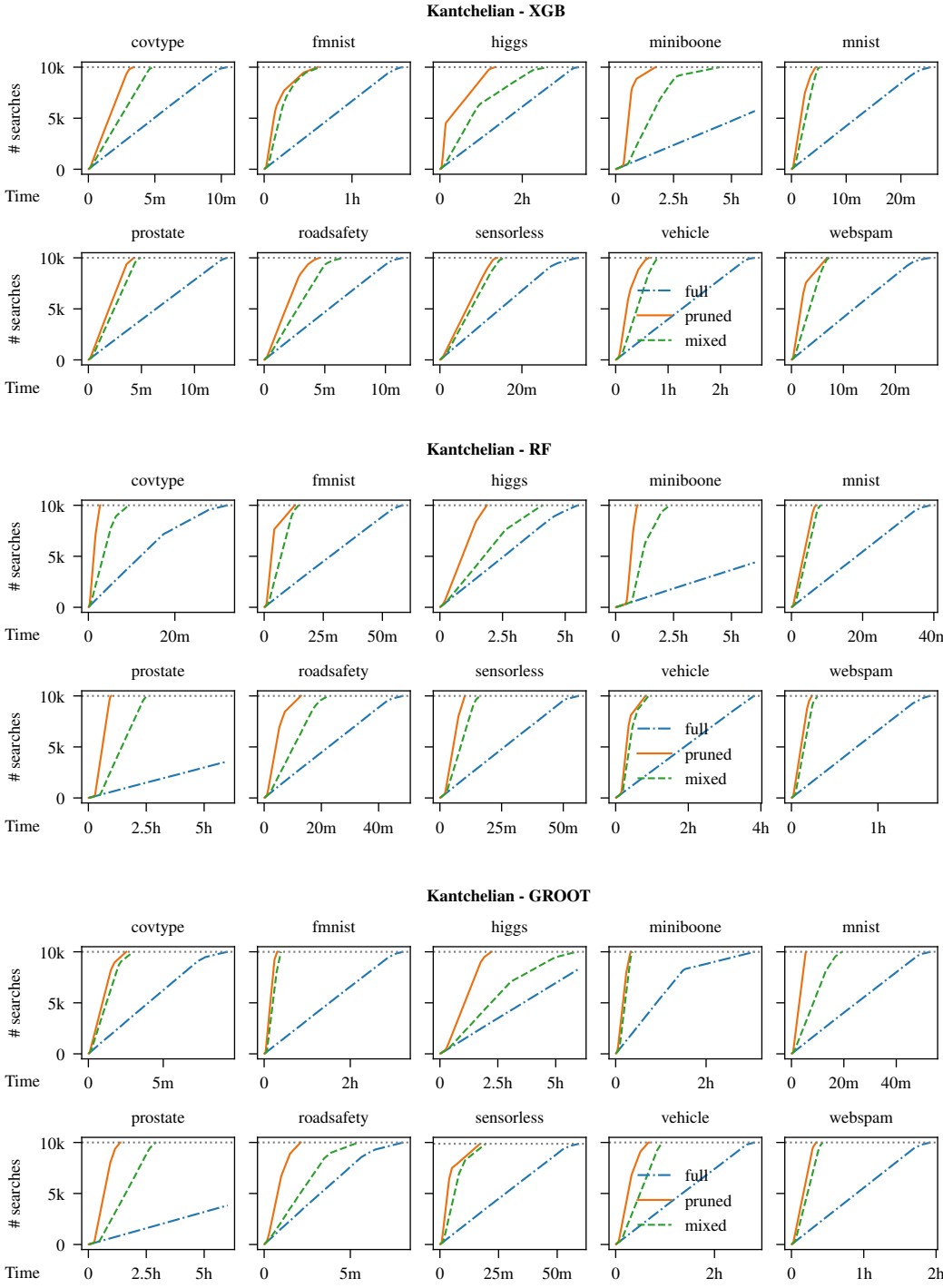

Figure 7: Run times to attempt to generate adversarial examples for 10 000 test examples with the three presented settings (*full*, *pruned* and *mixed*), using *kantchelian* on an XGBoost/random forest/GROOT forest ensemble, averaged over 5 folds.

In Figure 9, we show the empirical FNR on the x-axis versus the speedup of the *pruned* approach on the y-axis, using an XGB ensemble on four selected datasets. The empirical FNR is the fraction of times the *pruned* approach returns $UNSAT$ but the full approach returns $SAT$. Going from left to

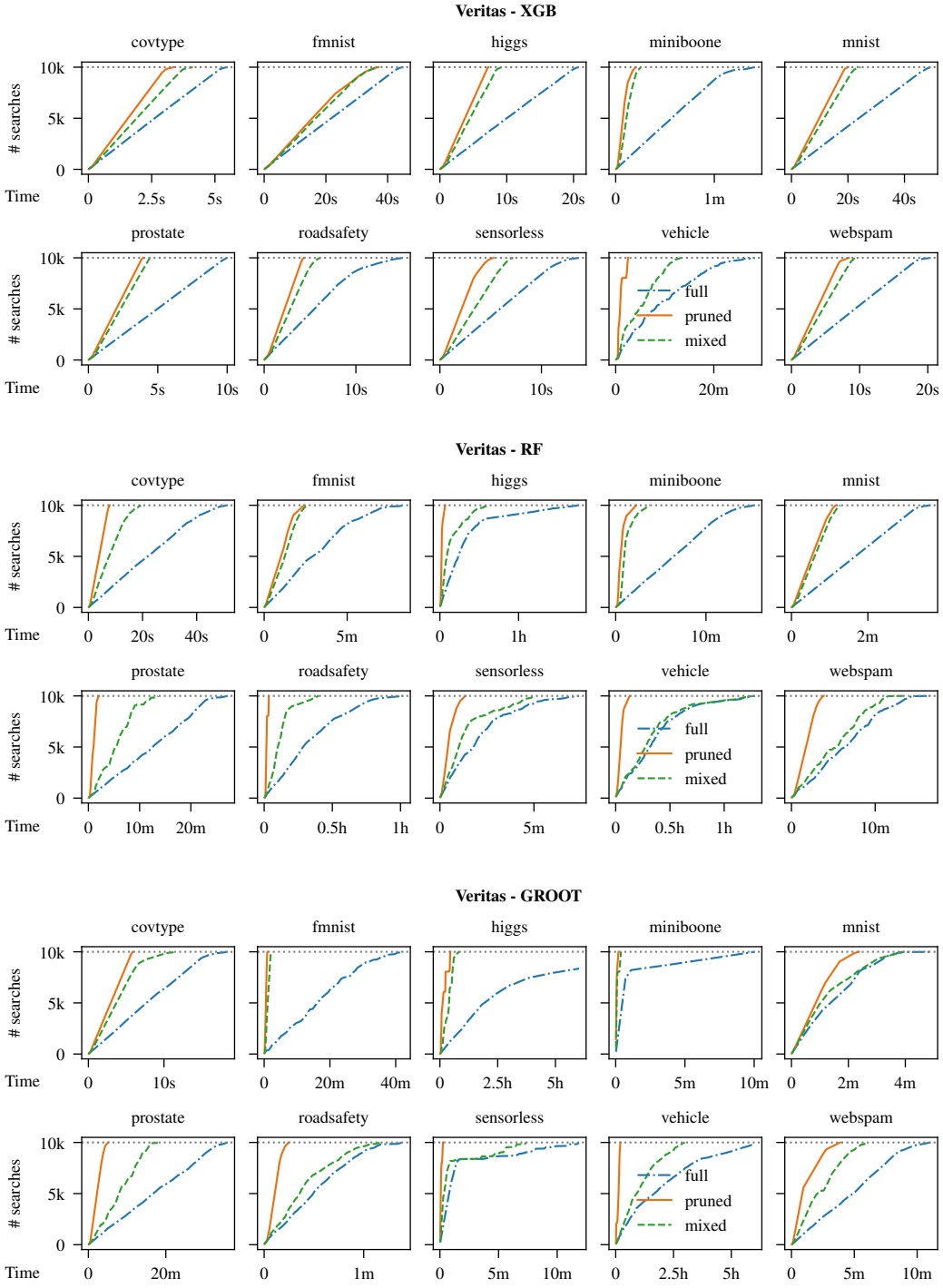

Figure 8: Run times to attempt to generate adversarial examples for 10 000 test examples with the three presented settings (*full*, *pruned* and *mixed*), using *veritas* on an XGBoost/random forest/GROOT forest ensemble, averaged over 5 folds.

right along the x-axis, higher values of the FNR correspond to smaller subsets of selected features, hence more aggressive pruning. Using less features, the *pruned* search becomes faster, at the cost of more false negatives.

Table 7: Average run times (with standard deviations) when attempting to generate 10 000 adversarial examples using *kantchelian* on an XGBoost/random forest/GROOT forest ensemble for all three approaches: *full*, *pruned* and *mixed*. The average fraction of timeouts incurred during the search is also reported.

**Kantchelian, XGBoost**

|  | run times | | | % timeouts | | |
|---|---|---|---|---|---|---|
|  | *full* | *pruned* | *mixed* | *full* | *pruned* | *mixed* |
| covtype | 9.9m ± 23.0s | 3.0m ± 12.3s | 4.7m ± 8.4s | 0% | 0% | 0% |
| fmnist | 1.5h ± 4.1m | 18.5m ± 11.4m | 22.9m ± 9.5m | 0% | <1% | 0% |
| higgs | 3.3h ± 3.3m | 47.7m ± 32.6m | 1.8h ± 42.7m | 0% | 9.6% | 0% |
| miniboone | 6.0h ± 1.8s | 55.5m ± 24.2m | 2.8h ± 52.3m | 0% | <1% | 0% |
| mnist | 23.9m ± 1.3m | 3.4m ± 52.1s | 4.7m ± 24.1s | 0% | 0% | 0% |
| prostate | 12.8m ± 14.3s | 3.8m ± 20.5s | 4.6m ± 11.0s | 0% | 0% | 0% |
| roadsafety | 10.7m ± 28.2s | 3.6m ± 40.9s | 5.4m ± 35.3s | 0% | 0% | 0% |
| sensorless | 29.8m ± 2.8m | 12.8m ± 58.0s | 14.4m ± 1.2m | <1% | 0% | 0% |
| vehicle | 2.5h ± 5.0m | 25.7m ± 9.2m | 44.2m ± 4.3m | 0% | <1% | 0% |
| webspam | 24.2m ± 1.7m | 4.3m ± 2.1m | 6.6m ± 35.4s | 0% | 0% | 0% |

**Kantchelian, RF**

|  | run times | | | % timeouts | | |
|---|---|---|---|---|---|---|
|  | *full* | *pruned* | *mixed* | *full* | *pruned* | *mixed* |
| covtype | 25.7m ± 6.2m | 2.3m ± 29.0s | 6.9m ± 1.9m | 0% | 0% | 0% |
| fmnist | 56.2m ± 2.3m | 7.2m ± 4.1m | 13.0m ± 1.2m | 0% | 0% | 0% |
| higgs | 5.1h ± 27.5m | 1.7h ± 12.6m | 3.6h ± 42.7m | 0% | 19.7% | 0% |
| miniboone | 6.0h ± 2.8s | 52.0m ± 4.8m | 1.9h ± 26.7m | 0% | 0% | 0% |
| mnist | 36.6m ± 1.8m | 6.4m ± 22.6s | 7.6m ± 22.7s | 0% | 0% | 0% |
| prostate | 6.0h ± 2.0s | 56.4m ± 1.6m | 2.5h ± 4.6m | <1% | <1% | 0% |
| roadsafety | 45.2m ± 2.6m | 8.4m ± 3.2m | 19.7m ± 2.2m | <1% | 0% | 0% |
| sensorless | 52.5m ± 2.7m | 9.2m ± 1.2m | 14.8m ± 1.0m | 0% | 0% | 0% |
| vehicle | 3.8h ± 2.5m | 31.8m ± 12.4m | 39.3m ± 11.2m | 0% | 0% | 0% |
| webspam | 1.5h ± 5.0m | 12.4m ± 1.4m | 15.9m ± 1.6m | 0% | 0% | 0% |

**Kantchelian, GROOT**

|  | run times | | | % timeouts | | |
|---|---|---|---|---|---|---|
|  | *full* | *pruned* | *mixed* | *full* | *pruned* | *mixed* |
| covtype | 8.1m ± 50.7s | 1.9m ± 27.2s | 2.4m ± 26.5s | 0% | 0% | 0% |
| fmnist | 3.1h ± 8.5m | 15.6m ± 1.8m | 21.4m ± 3.0m | 0% | 0% | 0% |
| higgs | 6.0h ± 0.9s | 2.0h ± 12.0m | 4.6h ± 1.2h | 0% | 18.9% | 0% |
| miniboone | 2.0h ± 44.7m | 17.8m ± 2.6m | 20.9m ± 1.2m | 0% | 0% | 0% |
| mnist | 50.0m ± 2.2m | 5.5m ± 3.7s | 16.3m ± 2.5m | 0% | 0% | 0% |
| prostate | 6.0h ± 0.9s | 1.2h ± 10.5m | 2.8h ± 7.3m | 0% | <1% | 0% |
| roadsafety | 6.7m ± 1.0m | 1.5m ± 28.2s | 4.2m ± 52.0s | 0% | 0% | 0% |
| sensorless | 55.6m ± 2.8m | 8.6m ± 6.0m | 12.7m ± 4.9m | 0% | 0% | 0% |
| vehicle | 2.7h ± 4.6m | 29.7m ± 8.4m | 53.8m ± 3.5m | <1% | <1% | 0% |
| webspam | 1.8h ± 5.7m | 18.6m ± 1.4m | 24.1m ± 1.6m | 0% | 0% | 0% |

There are cases where false negatives are less problematic, such as when one simply needs to generate a lot of adversarial examples for model hardening [23]. In these cases, the *pruned* approach really excels at offering run time improvements.

Note that for the considered datasets, FNR values higher than 25% are rare and only occur for very small subsets of features (e.g., 5 out of the 784 features in *mnist*).

## D    Sensitivity Analysis

We briefly discuss how sensitive our algorithm is to the choice of its hyperparameters, namely the threshold and confidence for the statistical test in the feature selection process (see 3.2) and the timeout for the *pruned* setting.

Table 8: Average run times (with standard deviations) when attempting to generate 10 000 adversarial examples using *veritas* on an XGBoost/random forest/GROOT forest ensemble for all three approaches: *full*, *pruned* and *mixed*. The average fraction of timeouts incurred during the search is also reported.

**Veritas, XGBoost**

| | run times | | | % timeouts | | |
|---|---|---|---|---|---|---|
| | *full* | *pruned* | *mixed* | *full* | *pruned* | *mixed* |
| covtype | $5.3s \pm 0.1s$ | $3.1s \pm 0.2s$ | $3.8s \pm 0.2s$ | 0% | 0% | 0% |
| fmnist | $43.6s \pm 1.1s$ | $31.9s \pm 5.4s$ | $33.6s \pm 3.0s$ | 0% | <1% | 0% |
| higgs | $20.4s \pm 0.3s$ | $7.1s \pm 0.1s$ | $8.6s \pm 0.4s$ | 0% | 0% | 0% |
| miniboone | $1.2m \pm 8.0s$ | $8.5s \pm 2.4s$ | $12.0s \pm 2.5s$ | 0% | 0% | 0% |
| mnist | $47.9s \pm 1.7s$ | $19.0s \pm 0.6s$ | $22.4s \pm 1.1s$ | 0% | 0% | 0% |
| prostate | $9.9s \pm 0.2s$ | $3.9s \pm 0.1s$ | $4.4s \pm 0.0s$ | 0% | 0% | 0% |
| roadsafety | $11.4s \pm 2.7s$ | $4.2s \pm 0.1s$ | $5.4s \pm 0.5s$ | 0% | 0% | 0% |
| sensorless | $12.1s \pm 1.1s$ | $4.2s \pm 0.8s$ | $6.6s \pm 0.5s$ | 0% | 0% | 0% |
| vehicle | $19.4m \pm 7.3m$ | $1.2m \pm 40.5s$ | $10.0m \pm 4.2m$ | <1% | <1% | <1% |
| webspam | $18.8s \pm 0.8s$ | $7.3s \pm 0.6s$ | $9.0s \pm 0.3s$ | 0% | 0% | 0% |

**Veritas, RF**

| | run times | | | % timeouts | | |
|---|---|---|---|---|---|---|
| | *full* | *pruned* | *mixed* | *full* | *pruned* | *mixed* |
| covtype | $45.0s \pm 5.5s$ | $7.5s \pm 0.3s$ | $15.4s \pm 2.8s$ | 0% | 0% | 0% |
| fmnist | $6.6m \pm 1.2m$ | $1.8m \pm 19.3s$ | $2.2m \pm 19.0s$ | <1% | 3.4% | <1% |
| higgs | $41.8m \pm 32.7m$ | $2.0m \pm 1.1m$ | $16.8m \pm 12.2m$ | <1% | 1.1% | <1% |
| miniboone | $12.9m \pm 1.6m$ | $1.1m \pm 38.2s$ | $2.0m \pm 1.0m$ | <1% | <1% | 0% |
| mnist | $3.3m \pm 9.5s$ | $1.1m \pm 5.7s$ | $1.2m \pm 5.8s$ | 0% | 1.3% | 0% |
| prostate | $23.7m \pm 1.7m$ | $1.4m \pm 28.2s$ | $9.3m \pm 2.4m$ | <1% | <1% | <1% |
| roadsafety | $40.6m \pm 14.7m$ | $1.2m \pm 28.1s$ | $13.0m \pm 5.6m$ | <1% | 0% | 0% |
| sensorless | $4.1m \pm 1.9m$ | $52.3s \pm 19.1s$ | $2.8m \pm 1.7m$ | 0% | <1% | 0% |
| vehicle | $42.8m \pm 18.9m$ | $4.7m \pm 1.6m$ | $38.9m \pm 19.7m$ | <1% | 15.6% | <1% |
| webspam | $12.9m \pm 2.5m$ | $3.3m \pm 27.4s$ | $10.6m \pm 2.1m$ | <1% | 11.0% | <1% |

**Veritas, GROOT**

| | run times | | | % timeouts | | |
|---|---|---|---|---|---|---|
| | *full* | *pruned* | *mixed* | *full* | *pruned* | *mixed* |
| covtype | $15.9s \pm 1.4s$ | $5.8s \pm 0.2s$ | $7.7s \pm 1.9s$ | 0% | 0% | 0% |
| fmnist | $32.2m \pm 8.4m$ | $53.4s \pm 4.4s$ | $1.6m \pm 23.2s$ | <1% | <1% | <1% |
| higgs | $4.1h \pm 1.7h$ | $11.6m \pm 8.7m$ | $31.5m \pm 13.0m$ | 1.1% | 4.0% | <1% |
| miniboone | $2.6m \pm 3.7m$ | $6.1s \pm 2.9s$ | $9.7s \pm 6.5s$ | 0% | 0% | 0% |
| mnist | $3.1m \pm 59.0s$ | $1.8m \pm 25.4s$ | $2.5m \pm 54.8s$ | <1% | 4.1% | <1% |
| prostate | $33.9m \pm 2.1m$ | $4.2m \pm 36.4s$ | $15.7m \pm 1.2m$ | <1% | 12.3% | <1% |
| roadsafety | $1.0m \pm 16.8s$ | $11.4s \pm 2.1s$ | $48.6s \pm 18.5s$ | 0% | 0% | 0% |
| sensorless | $3.4m \pm 4.2m$ | $8.6s \pm 3.6s$ | $1.9m \pm 2.7m$ | <1% | 0% | 0% |
| vehicle | $4.0h \pm 1.9h$ | $8.2m \pm 3.7m$ | $2.1h \pm 50.4m$ | 1.2% | 10.7% | <1% |
| webspam | $9.5m \pm 51.9s$ | $2.2m \pm 1.1m$ | $4.6m \pm 1.3m$ | <1% | 6.6% | <1% |

## D.1 Sensitivity to Statistical Test Parameters

The statistical test described in 3.2 takes as hyperparameters the acceptable false negative rate $\tau$ and the confidence $1 - \eta$. We perform a sensitivity analysis where we vary $\tau \in [0.05, 0.1, 0.25, 0.5]$ and $1 - \eta \in [0.8, 0.9, 0.95]$ on the *miniboone* dataset.

Table 9 shows the *mixed* speedup for (*veritas*, XGBoost) for all combinations of the considered values for $\tau$ (FNR) and $1 - \eta$ (confidence). For all settings, our approach improves upon the run time of always running a *full* search (i.e., speedup is always $> 1$). When $\tau = 0.05$, many features are selected and hence there is less pruning. $\tau = 0.1$ and $\tau = 0.25$ perform identically. When $\tau = 0.5$, the value of $1 - \eta$ impacts the selected feature set. A confidence of 0.95 keeps the same feature set of $\tau = 0.1$ and $\tau = 0.25$. However, a lower confidence results in an even smaller feature set, which degrades the performance of the *mixed* setting because there are more calls to the full search.

Hence more in general, the threshold on the false negative rate $\tau$ is inversely proportional to the number of chosen features: the larger the selected feature subset, the lower the FNR will be. This is in tension with the goal of using as small of a feature subset as possible, to speed up the *pruned*

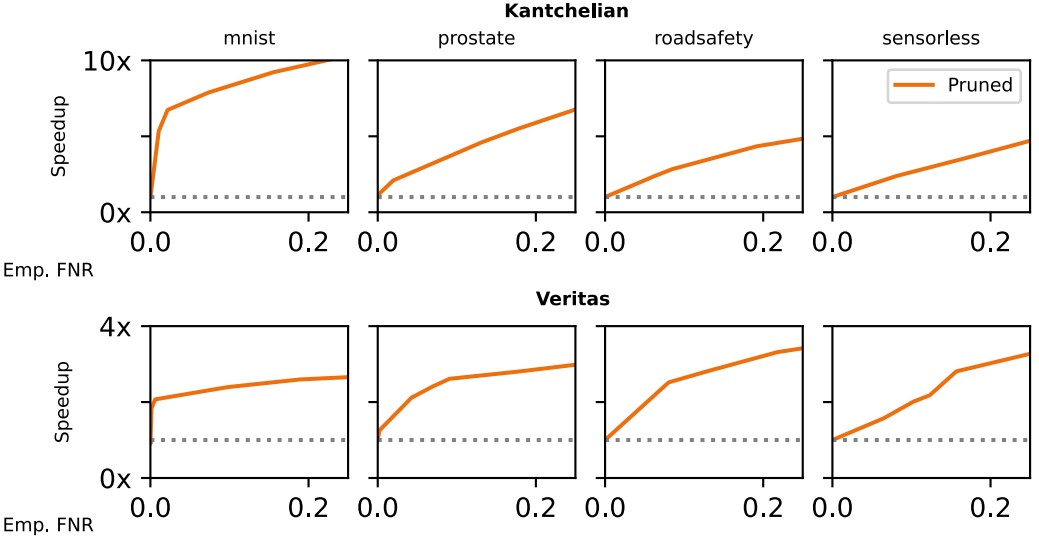

Figure 9: Speedups introduced by the *pruned* setting when attempting to generate 10 000 test examples using *kantchelian* (top) and *veritas* (bottom) on an XGB ensemble, varying the empirical false negative rate. Higher FNRs correspond to smaller feature subsets. The dotted horizontal line corresponds to a speedup of 1x, i.e., same run time of the *full* setting.

setting. Moreover, a larger confidence $1 - \eta$ can increase the number of selected features as it shrinks the confidence interval for the empirical FNR.

Table 9: Speedup of the *mixed* setting when attempting to generate 10 000 adversarial examples for *miniboone* using (*veritas*, XGBoost), for different values of $\tau$ (threshold on the allowed false negative rate) and $1 - \eta$ (confidence of the statistical test).

| $\tau$ | $1 - \eta$ | *mixed* speedup |
|---|---|---|
| 0.05 | 0.8 | 3.7x |
| | 0.9 | 3.7x |
| | 0.95 | 3.7x |
| 0.10 | 0.8 | 6.3x |
| | 0.9 | 6.3x |
| | 0.95 | 6.3x |
| 0.25 | 0.8 | 6.3x |
| | 0.9 | 6.3x |
| | 0.95 | 6.3x |
| 0.50 | 0.8 | 1.9x |
| | 0.9 | 1.9x |
| | 0.95 | 6.3x |

## D.2 Sensitivity to Timeouts

Timeouts always need to be explicitly handled, due to the hardness of the evasion problem [23]. While Tables 7 and 8 show that in most of our experiments timeouts are rare or totally absent, to complete the discussion we perform a sensitivity analysis on the value used for the *pruned* setting timeout. We vary $t_{\max}^{prun} \in [0.001, 0.01, 0.1, 1]$ seconds when attempting to generate 10 000 adversarial examples on the *fmnist* dataset. We only consider *veritas* for RF in this experiment.

Table 10 shows the fraction of timeouts for the *pruned* setting as well as the *pruned* speedup and the *mixed* speedup for each value of $t_{\max}^{prun}$. The smaller $t_{\max}^{prun}$, the larger the number *pruned* timeouts,

which corresponds to a faster (but less accurate) *pruned* search. If the timeout value is too low, this adversely affects *mixed* search because it leads to more calls to the *full* procedure. Conversely, if $t_{\max}^{prun}$ is too large, the search becomes slower as the *pruned* setting starts losing time on a few slow instances.

Thus the ideal $t_{\max}^{prun}$ lays in between the two extremes. In the presented case, $t_{\max}^{prun} = 0.01$ works best. The best choice likely depends on the specific dataset, model, and attack type. However, tuning its value is time consuming (i.e., negates the benefits of the proposed approach).

Table 10: Fraction of *pruned* timeouts and speedup of the *pruned* and *mixed* settings when attempting to generate 10 000 adversarial examples for *fmnist* using (*veritas*, random forest), for different values of the *pruned* setting timeout $t_{\max}^{prun}$ (in seconds).

| $t_{\max}^{prun}$ (s) | % *pruned* timeouts | *pruned* speedup | *mixed* speedup |
|---|---|---|---|
| 0.001 | 24% | 7.9× | 2.8× |
| 0.01 | 12% | 5.8× | 3.2× |
| 0.1 | 4% | 2.8× | 2.3× |
| 1 | 1% | 1.1× | 1.1× |

# E  Quality of Generated Adversarial Examples

We extend Figure 4 by further discussing the quality of generated adversarial examples, providing more examples, and looking in detail at their distance with respect to the base examples.

Figure 10 shows a large set of adversarial examples generated for *mnist* digits using *kantchelian* and *veritas* on an XGBoost ensemble. For each attack, we plot the base example $x$ and the two adversarial examples generated with the *full* and the *pruned* setting.

## E.1  Empirical Robustness

Tables 11 (*kantchelian*) and 12 (*veritas*) show the average empirical robustness in all the performed experiments for the *full*, *pruned* and *mixed* settings. An ensemble's *empirical robustness* is defined as the average distance to the nearest adversarial example for each $x$ in a test set. We use adversarial examples generated with the experiments presented in Q1 in Section 4 (and Appendix C).

The objective of the *kantchelian* attack is to find the closest adversarial example. Given that the method is exact, the *full* setting returns the optimal solution. The *pruned* search works with a restricted feature set, thus it might not be able to find the closest adversarial example, if that requires altering features not included in the selected feature subset. As a consequence, the *empirical robustness* values for the *pruned* and *mixed* search are overestimates of the true value given by the *full* setting.

Unlike *kantchelian*, *veritas* does not try to find the closest adversarial example. Instead, it maximizes the confidence that the ensemble assigns to the incorrect label. In this case, there is little difference in the empirical robustness values among all considered settings, with the *pruned* and *mixed* settings typically managing to even lower the distance to the base example.

## E.2  Change in Predicted Probability for Adversarial Examples

*veritas* tries to generate an adversarial example such that the ensemble assigns as high a probability as possible to the incorrect label. Hence, a natural empirical measure for the quality of the generated examples is to compare the difference in the ensembles probabilistic predictions for the adversarial examples generated by each approach. Namely, we compute $\boldsymbol{T}(\tilde{x})$ - $\boldsymbol{T}(\tilde{x}')$ where $\tilde{x}$ is generated by the full search, $\tilde{x}'$ is generated by the *pruned* (*mixed*) search, and (in an abuse of notation) $\boldsymbol{T}(x)$ returns the probability an example belongs to most likely class.

Table 13 shows the average differences in predicted probability between *full* and *pruned/mixed* adversarial examples.

Using *kantchelian*, adversarial examples generated with our approaches are assigned very similar probabilities to those generated with the *full* search. In *veritas*, differences are typically higher, as the model output is directly optimized.

Table 11: Average empirical robustness (i.e., distance to the closest adversarial example) for the *full*, *mixed* and *pruned* methods using *kantchelian* attack on XGBoost/random forest/GROOT forest ensembles.

**Kantchelian, XGBoost**

|  | *full* | *pruned* | *mixed* |
|---|---|---|---|
| covtype | $0.018 \pm 0.0$ | $0.038 \pm 0.0$ | $0.038 \pm 0.0$ |
| fmnist | $0.034 \pm 0.002$ | $0.056 \pm 0.012$ | $0.056 \pm 0.012$ |
| higgs | $0.011 \pm 0.0$ | $0.015 \pm 0.002$ | $0.016 \pm 0.002$ |
| miniboone | $0.001 \pm 0.0$ | $0.001 \pm 0.0$ | $0.001 \pm 0.0$ |
| mnist | $0.006 \pm 0.001$ | $0.021 \pm 0.007$ | $0.02 \pm 0.007$ |
| prostate | $0.02 \pm 0.0$ | $0.037 \pm 0.001$ | $0.038 \pm 0.001$ |
| roadsafety | $0.005 \pm 0.0$ | $0.014 \pm 0.003$ | $0.015 \pm 0.003$ |
| sensorless | $0.006 \pm 0.001$ | $0.008 \pm 0.001$ | $0.009 \pm 0.0$ |
| vehicle | $0.017 \pm 0.001$ | $0.045 \pm 0.013$ | $0.042 \pm 0.011$ |
| webspam | $0.002 \pm 0.0$ | $0.006 \pm 0.002$ | $0.006 \pm 0.002$ |

**Kantchelian, RF**

|  | *full* | *pruned* | *mixed* |
|---|---|---|---|
| covtype | $0.102 \pm 0.003$ | $0.113 \pm 0.002$ | $0.117 \pm 0.001$ |
| fmnist | $0.02 \pm 0.002$ | $0.046 \pm 0.013$ | $0.045 \pm 0.012$ |
| higgs | $0.016 \pm 0.0$ | $0.016 \pm 0.001$ | $0.019 \pm 0.0$ |
| miniboone | $0.001 \pm 0.0$ | $0.001 \pm 0.0$ | $0.001 \pm 0.0$ |
| mnist | $0.006 \pm 0.001$ | $0.023 \pm 0.002$ | $0.023 \pm 0.002$ |
| prostate | $0.058 \pm 0.0$ | $0.089 \pm 0.0$ | $0.094 \pm 0.0$ |
| roadsafety | $0.022 \pm 0.001$ | $0.025 \pm 0.004$ | $0.028 \pm 0.003$ |
| sensorless | $0.017 \pm 0.002$ | $0.032 \pm 0.001$ | $0.032 \pm 0.001$ |
| vehicle | $0.015 \pm 0.001$ | $0.033 \pm 0.006$ | $0.033 \pm 0.006$ |
| webspam | $0.003 \pm 0.0$ | $0.006 \pm 0.001$ | $0.006 \pm 0.001$ |

**Kantchelian, GROOT**

|  | *full* | *pruned* | *mixed* |
|---|---|---|---|
| covtype | $0.124 \pm 0.001$ | $0.136 \pm 0.002$ | $0.139 \pm 0.002$ |
| fmnist | $0.295 \pm 0.001$ | $0.309 \pm 0.002$ | $0.309 \pm 0.002$ |
| higgs | $0.109 \pm 0.008$ | $0.122 \pm 0.015$ | $0.127 \pm 0.01$ |
| miniboone | $0.025 \pm 0.0$ | $0.048 \pm 0.014$ | $0.048 \pm 0.014$ |
| mnist | $0.277 \pm 0.001$ | $0.296 \pm 0.001$ | $0.298 \pm 0.0$ |
| prostate | $0.063 \pm 0.0$ | $0.091 \pm 0.003$ | $0.096 \pm 0.003$ |
| roadsafety | $0.092 \pm 0.006$ | $0.075 \pm 0.011$ | $0.092 \pm 0.007$ |
| sensorless | $0.051 \pm 0.003$ | $0.077 \pm 0.013$ | $0.079 \pm 0.013$ |
| vehicle | $0.149 \pm 0.001$ | $0.165 \pm 0.003$ | $0.168 \pm 0.002$ |
| webspam | $0.034 \pm 0.001$ | $0.052 \pm 0.002$ | $0.052 \pm 0.002$ |

## F   Expanded Related Work

Adversarial examples have been theoretically studied and defined in multiple different ways [14, 18]. More specifically, Ilyas et al. showed how certain features in a dataset might be fragile and thus naturally lead to adversarial examples [21]. Approaches to reason about learned tree ensembles have received substantial interest in recent years. These include algorithms for performing evasion attacks [23, 15] (i.e., generate adversarial examples), perform robustness checking [8], and verify that the ensembles satisfy certain criteria [11, 10, 26, 29]. Kantchelian et al. [23] were the first to show that, just like neural networks, tree ensembles are susceptible to evasion attacks. Their MILP formulation is still the most frequently used method to check robustness and generate adversarial examples. Other notable methods for adversarial example generation are SMT-based systems [15, 11]. These approaches propose varying ways to encode a tree ensemble in a set of logical formulas using the primitives from Satisfiability Modulo Theories (SMT). While the formulation of an ensemble in SMT is very elegant, it tends to perform worse than MILP in practice.

Because MILP and SMT are exact approaches,[6] they search for the optimal answer which in certain cases can be difficult (i.e., time consuming) to find. Often an approximate answer will be sufficient and several approximate methods have been proposed that are specifically tailored to tree ensembles. Chen et al. proposed a $K$-partite graph representation in which a max-clique corresponds to a specific output of the ensemble [8, 35]. They introduced a fast method to approximately evaluate robustness,

---

[6]MILP is technically anytime, but the approximate solutions are not useful in practice for this problem setting, see [10].

Table 12: Average empirical robustness (i.e., distance to the closest adversarial example) for the *full*, *mixed* and *pruned* methods using *veritas* attack on XGBoost/random forest/GROOT forest ensembles.

**Veritas, XGBoost**

|            | *full*          | *pruned*          | *mixed*           |
|------------|-----------------|-------------------|-------------------|
| covtype    | $0.094 \pm 0.0$ | $0.088 \pm 0.001$ | $0.088 \pm 0.001$ |
| fmnist     | $0.287 \pm 0.002$ | $0.275 \pm 0.004$ | $0.276 \pm 0.004$ |
| higgs      | $0.071 \pm 0.0$ | $0.061 \pm 0.003$ | $0.061 \pm 0.003$ |
| miniboone  | $0.06 \pm 0.003$ | $0.026 \pm 0.013$ | $0.029 \pm 0.012$ |
| mnist      | $0.289 \pm 0.003$ | $0.253 \pm 0.016$ | $0.257 \pm 0.013$ |
| prostate   | $0.097 \pm 0.0$ | $0.095 \pm 0.0$   | $0.095 \pm 0.0$   |
| roadsafety | $0.057 \pm 0.0$ | $0.055 \pm 0.0$   | $0.056 \pm 0.0$   |
| sensorless | $0.055 \pm 0.0$ | $0.047 \pm 0.006$ | $0.049 \pm 0.004$ |
| vehicle    | $0.14 \pm 0.0$  | $0.133 \pm 0.003$ | $0.134 \pm 0.002$ |
| webspam    | $0.038 \pm 0.0$ | $0.035 \pm 0.001$ | $0.035 \pm 0.001$ |

**Veritas, RF**

|            | *full*          | *pruned*          | *mixed*           |
|------------|-----------------|-------------------|-------------------|
| covtype    | $0.272 \pm 0.001$ | $0.249 \pm 0.006$ | $0.251 \pm 0.006$ |
| fmnist     | $0.293 \pm 0.0$ | $0.277 \pm 0.004$ | $0.278 \pm 0.004$ |
| higgs      | $0.07 \pm 0.001$ | $0.065 \pm 0.002$ | $0.066 \pm 0.002$ |
| miniboone  | $0.061 \pm 0.002$ | $0.038 \pm 0.01$ | $0.04 \pm 0.01$   |
| mnist      | $0.289 \pm 0.002$ | $0.275 \pm 0.004$ | $0.276 \pm 0.004$ |
| prostate   | $0.194 \pm 0.0$ | $0.186 \pm 0.002$ | $0.188 \pm 0.002$ |
| roadsafety | $0.11 \pm 0.001$ | $0.094 \pm 0.006$ | $0.097 \pm 0.005$ |
| sensorless | $0.113 \pm 0.0$ | $0.101 \pm 0.004$ | $0.103 \pm 0.003$ |
| vehicle    | $0.138 \pm 0.0$ | $0.129 \pm 0.004$ | $0.134 \pm 0.003$ |
| webspam    | $0.058 \pm 0.0$ | $0.054 \pm 0.001$ | $0.055 \pm 0.001$ |

**Veritas, GROOT**

|            | *full*          | *pruned*          | *mixed*           |
|------------|-----------------|-------------------|-------------------|
| covtype    | $0.359 \pm 0.002$ | $0.329 \pm 0.006$ | $0.331 \pm 0.006$ |
| fmnist     | $0.394 \pm 0.0$ | $0.382 \pm 0.003$ | $0.383 \pm 0.003$ |
| higgs      | $0.37 \pm 0.002$ | $0.354 \pm 0.008$ | $0.357 \pm 0.007$ |
| miniboone  | $0.459 \pm 0.047$ | $0.223 \pm 0.049$ | $0.233 \pm 0.051$ |
| mnist      | $0.397 \pm 0.0$ | $0.389 \pm 0.003$ | $0.39 \pm 0.003$  |
| prostate   | $0.196 \pm 0.0$ | $0.19 \pm 0.001$  | $0.192 \pm 0.0$   |
| roadsafety | $0.191 \pm 0.001$ | $0.187 \pm 0.001$ | $0.188 \pm 0.001$ |
| sensorless | $0.189 \pm 0.0$ | $0.166 \pm 0.009$ | $0.17 \pm 0.007$  |
| vehicle    | $0.392 \pm 0.0$ | $0.378 \pm 0.01$  | $0.382 \pm 0.007$ |
| webspam    | $0.098 \pm 0.0$ | $0.096 \pm 0.001$ | $0.097 \pm 0.0$   |

but it cannot generate concrete adversarial examples. Devos et al. further improved upon this work by proposing a heuristic search procedure in this graph which is capable of finding concrete adversarial examples very effectively [10]. Zhang et al. propose a method based on a greedy discrete search through the space of leaves specifically optimized for fast adversarial example generation [36].

Other work focuses on making tree ensembles more robust. There are multiple approaches: adding generated adversarial examples to the training data (model hardening) [23], modifying the splitting procedure [7, 4, 32], using the framework of optimal decision trees to encode robustness constraints [33], relabeling and pruning the leaves of the trees [34], simplifying the base learner [1] and using a robust 0/1 loss [19]. Gaining further insights into how evasion attacks target tree ensembles, like those contained in this paper, may inspire novel ways to improve the robustness of learners.

Another line of work aims at directly training tree ensembles that admit verification in polynomial time [5, 13]. However, a drawback to current approaches is that they result in (large) decreases in predictive performance.

Finally, performing evasion attacks has been studied for other model classes with deep neural networks receiving particular attention [28, 17, 24, 6]. However, state-of-the-art algorithms are tailored to one specific model type as they typically exploit specific properties of the model, e.g., the work on tree ensembles often exploits the logical structure of a decision tree.

Table 13: Average difference in predicted probability between an adversarial example generated using *kantchelian/veritas* with the *full* setting and an adversarial example generated with the *pruned/mixed* setting, for the same base example. All adversarial examples are those generated during the experiments from Section 4 and Appendix C.

**Kantchelian**

| | XGBoost | | RF | | GROOT | |
|---|---|---|---|---|---|---|
| | *pruned* | *mixed* | *pruned* | *mixed* | *pruned* | *mixed* |
| covtype | $0.093 \pm 0.001$ | $0.082 \pm 0.002$ | $0.01 \pm 0.002$ | $0.009 \pm 0.002$ | $0.011 \pm 0.001$ | $0.011 \pm 0.001$ |
| fmnist | $0.023 \pm 0.004$ | $0.021 \pm 0.003$ | $0.012 \pm 0.001$ | $0.011 \pm 0.0$ | $0.019 \pm 0.002$ | $0.018 \pm 0.002$ |
| higgs | $0.012 \pm 0.001$ | $0.01 \pm 0.001$ | $0.005 \pm 0.0$ | $0.003 \pm 0.0$ | $0.007 \pm 0.002$ | $0.005 \pm 0.003$ |
| miniboone | $0.013 \pm 0.001$ | $0.011 \pm 0.001$ | $0.006 \pm 0.0$ | $0.006 \pm 0.0$ | $0.005 \pm 0.0$ | $0.005 \pm 0.0$ |
| mnist | $0.109 \pm 0.011$ | $0.103 \pm 0.011$ | $0.02 \pm 0.002$ | $0.019 \pm 0.002$ | $0.018 \pm 0.004$ | $0.016 \pm 0.004$ |
| prostate | $0.026 \pm 0.0$ | $0.023 \pm 0.0$ | $0.004 \pm 0.0$ | $0.003 \pm 0.0$ | $0.003 \pm 0.0$ | $0.003 \pm 0.0$ |
| roadsafety | $0.135 \pm 0.025$ | $0.115 \pm 0.019$ | $0.013 \pm 0.002$ | $0.011 \pm 0.001$ | $0.033 \pm 0.016$ | $0.027 \pm 0.014$ |
| sensorless | $0.045 \pm 0.001$ | $0.041 \pm 0.002$ | $0.008 \pm 0.0$ | $0.007 \pm 0.0$ | $0.009 \pm 0.001$ | $0.008 \pm 0.001$ |
| vehicle | $0.015 \pm 0.004$ | $0.013 \pm 0.003$ | $0.005 \pm 0.0$ | $0.005 \pm 0.0$ | $0.005 \pm 0.001$ | $0.004 \pm 0.001$ |
| webspam | $0.049 \pm 0.006$ | $0.044 \pm 0.004$ | $0.007 \pm 0.0$ | $0.006 \pm 0.0$ | $0.006 \pm 0.0$ | $0.005 \pm 0.0$ |

**Veritas**

| | XGBoost | | RF | | GROOT | |
|---|---|---|---|---|---|---|
| | *pruned* | *mixed* | *pruned* | *mixed* | *pruned* | *mixed* |
| covtype | $0.129 \pm 0.02$ | $0.112 \pm 0.012$ | $0.069 \pm 0.007$ | $0.065 \pm 0.006$ | $0.054 \pm 0.005$ | $0.051 \pm 0.005$ |
| fmnist | $0.228 \pm 0.065$ | $0.213 \pm 0.056$ | $0.373 \pm 0.011$ | $0.327 \pm 0.009$ | $0.379 \pm 0.009$ | $0.367 \pm 0.01$ |
| higgs | $0.071 \pm 0.006$ | $0.067 \pm 0.005$ | $0.062 \pm 0.009$ | $0.054 \pm 0.007$ | $0.057 \pm 0.01$ | $0.051 \pm 0.009$ |
| miniboone | $0.246 \pm 0.042$ | $0.231 \pm 0.036$ | $0.178 \pm 0.027$ | $0.164 \pm 0.028$ | $0.104 \pm 0.022$ | $0.1 \pm 0.019$ |
| mnist | $0.196 \pm 0.026$ | $0.179 \pm 0.022$ | $0.294 \pm 0.012$ | $0.275 \pm 0.009$ | $0.241 \pm 0.022$ | $0.21 \pm 0.021$ |
| prostate | $0.237 \pm 0.01$ | $0.218 \pm 0.009$ | $0.225 \pm 0.024$ | $0.195 \pm 0.016$ | $0.214 \pm 0.006$ | $0.169 \pm 0.006$ |
| roadsafety | $0.17 \pm 0.04$ | $0.146 \pm 0.031$ | $0.083 \pm 0.009$ | $0.071 \pm 0.009$ | $0.059 \pm 0.006$ | $0.048 \pm 0.003$ |
| sensorless | $0.142 \pm 0.053$ | $0.116 \pm 0.039$ | $0.143 \pm 0.021$ | $0.122 \pm 0.013$ | $0.13 \pm 0.012$ | $0.113 \pm 0.01$ |
| vehicle | $0.21 \pm 0.026$ | $0.175 \pm 0.017$ | $0.207 \pm 0.048$ | $0.135 \pm 0.036$ | $0.087 \pm 0.024$ | $0.065 \pm 0.018$ |
| webspam | $0.274 \pm 0.008$ | $0.244 \pm 0.009$ | $0.267 \pm 0.013$ | $0.226 \pm 0.01$ | $0.276 \pm 0.023$ | $0.231 \pm 0.016$ |

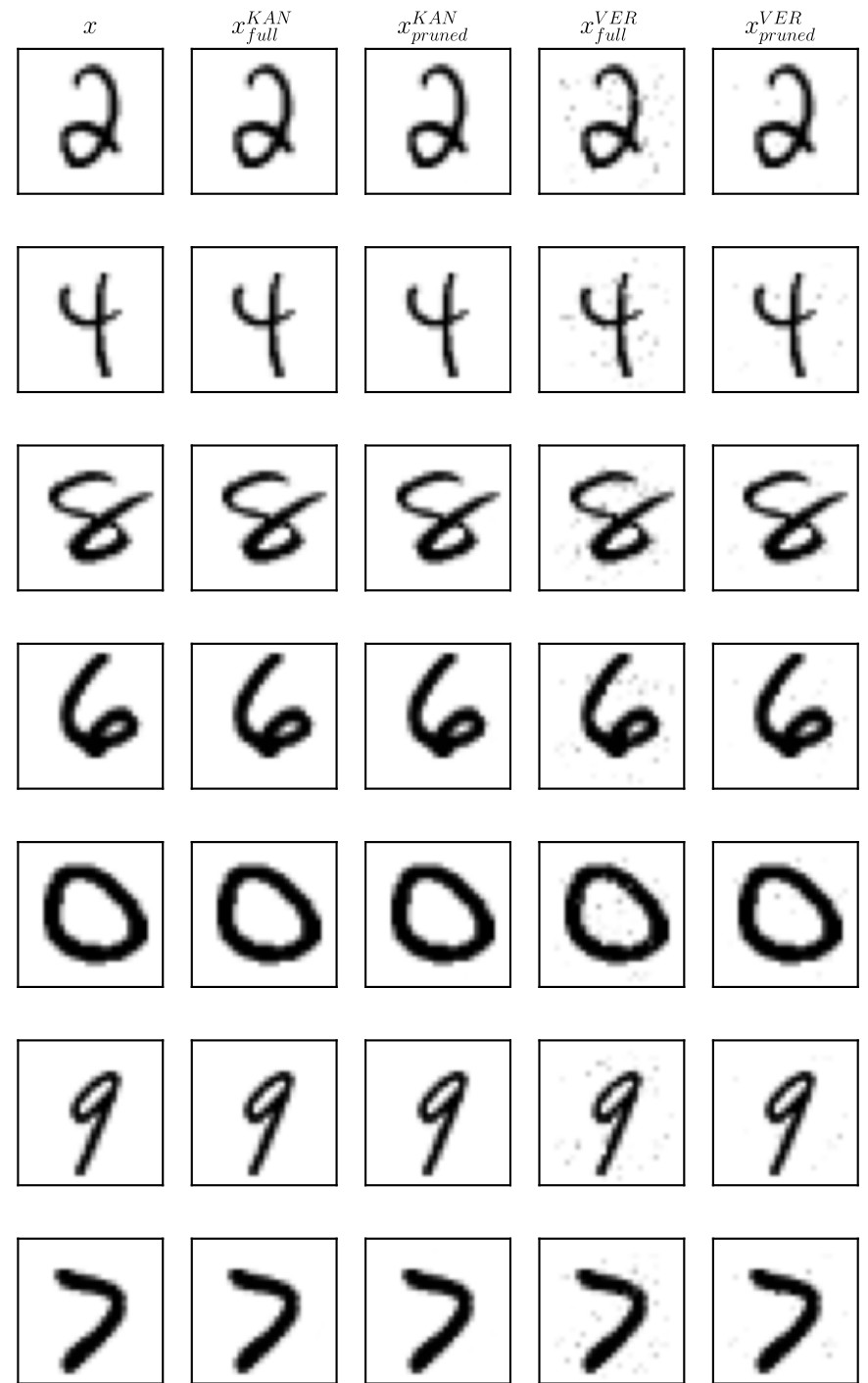

Figure 10: Adversarial examples generated for *mnist* with both attacks (*kantchelian* and *veritas*) on an XGBoost ensemble, to show the quality of generated examples.

