# OpenReview forum: "Faster Repeated Evasion Attacks in Tree Ensembles"
_NeurIPS.cc/2024/Conference — NeurIPS 2024 poster_

### Official Review · Reviewer_fYgE · 2024-06-30

**Soundness:** 3
**Presentation:** 3
**Contribution:** 3
**Rating:** 6
**Confidence:** 4

**Summary:**

The paper proposes a method to speed up the robustness verification of tree-based classifiers for all examples in a dataset. Previous methods solve the robustness verification problem for tree-based classifiers, which is NP-Complete, for each instance separately. However, the paper highlights that finding adversarial examples from a set of instances of the same dataset often requires perturbing only a small subset of features, called relevant features. The paper proposes a verification algorithm that employs state-of-the-art sound and complete verification algorithms. First, it tries to generate the adversarial example from this subset of relevant features and, if it is not found, tries again, considering all the features. Moreover, the paper proposes an algorithm to find the set of relevant features such that the probability that generating an adversarial example by perturbing only those features fails, even though the adversarial example exists, is less than a threshold with a certain confidence. The experimental evaluation considers two state-of-the-art verifiers and 11 tabular datasets commonly used in robustness verification literature. The experimental results show that verifying robustness by considering only the subset of relevant features results in a speed-up of up to 12x for gradient-boosted models and 6x for Random Forests.

**Strengths:**

**Original observation and algorithm that allow existing verifiers to speed up robustness verification of tree-based classifiers**: The observation that generating adversarial examples often requires perturbing only a subset of relevant features across instances of the same dataset is novel and leads to optimizing existing robustness verification algorithms. A disciplined approach is also provided to identify the subset of relevant features.

**Theoretical analysis of the guarantees provided by the proposed verification algorithm**: Two theorems and their proofs are provided regarding the guarantees of the verification algorithm: (i) if an adversarial example is found by perturbing only the subset of relevant features, the same adversarial example will also work when considering all the features; (ii) if an adversarial example exists, the algorithm is guaranteed to find it.

**The experimental settings considered are comprehensive and the results are significant**: The experimental evaluation is performed on several datasets and for two state-of-the-art verifiers. The results convincingly demonstrate the speed-up that can be achieved by considering only the relevant features. This is a significant result, as existing state-of-the-art verifiers struggle to verify the robustness of models based on decision trees.

**Weaknesses:**

**Only one threat model is considered**: Even though the $\ell_\infty$​ threat model is widely addressed in the literature, it would have been interesting to also consider the generalization of the proposal to other attackers like $\ell_1$​ and $\ell_2$ attackers that are considered in the literature when considering tabular data [1]. The kantchelian verifier can be used, since it supports also $\ell_1$​ and $\ell_2$ attackers.

**Missing comparison with other methods to improve the efficiency of robustness verification**: Proposals in the literature have addressed the problem of speeding up robustness verification and adversarial examples generation by training models that are amenable to robustness verification. First, neural networks have been considered [2], but recently, tree-based classifiers that admit robustness verification in polytime have also been explored [3]. The authors should consider comparing their methodology with [3], which is not addressed in the related work. They should highlight the pros and cons of their proposals with respect to this other way to make robustness verification of forests more efficient.

[1] Simonetto et. al., "A unified framework for adversarial attack and defense in constrained feature space", in IJCAI 2022.

[2] Xiao et. al., “Training for faster adversarial robustness verification via inducing relu stability,” in ICLR, 2019.

[3] Calzavara et. al., "Verifiable Learning for Robust Tree Ensembles", in CCS 2023.

**Questions:**

Is a set of relevant features present when generating adversarial examples against other threat models like $\ell_1$​ and $\ell_2$​? In other terms, does the proposed approach determine a speed-up when considering other attackers beyond $\ell_\infty$?

How does the proposed approach compare to approaches that speed up the efficiency of robustness verification by exploiting other points of view, like training models amenable to robustness verification [3]? (See the Weaknesses section for more details.)

**Limitations:**

The authors have explained the limitations of their approach appropriately, as claimed in the paper checklist. Furthermore, the authors have also discussed the potential negative societal impact of their work in Section 6. The experimental and implementation details have been thoroughly documented.

---

> ### Author Rebuttal · Authors · 2024-08-07
>
> **Why l-inf only?**
> We choose to work with the l-inf norm as recent works on approximate evasion
> attacks primarily focus on this scenario.
> This favours the experimental evaluation as these methods are the most efficient in literature, while still working in a perfectly reasonable scenario where the attack is defined by the magnitude of the largest perturbation. At the same time, we believe other attack models should present the same advantages when exploiting our approach, as the number of considered features is always a proxy for the hardness of the evasion.
>
> **Missing comparisons**
> We thank the reviewer for the multiple useful references which will include and discuss in the paper!  Ensuring polynomial verifiability is an exciting and important line of work. A drawback to current approaches for polynomial verifiability is that they clearly do result in (large) decreases in predictive performance. Whether this is permissible will depend on the considered application.  We would also highlight that Section C in the supplement shows results for applying our approach to GROOT forests which are robustified ensembles (Vos \& Verwer ICML'21).

---

> > ### Comment · Reviewer_fYgE · 2024-08-08
> > **Thanks for your response**
> >
> > I thank the authors for their response.
> >
> > It would have been interesting to see some results on your approach applied with $l_1$ or $l_2$-norm attackers, but I understand that computing these new results requires time. Additionally, since Veritas supports only $l_\infty$-attackers, this tool would have required an extension. However, I agree with your statement that the proposed approach should show advantages even when applied to these two threat models. I hope you will provide some experimental evidence on this fact in a next version of the paper.
> >
> > I strongly encourage the authors to add the following to the paper:
> > - Acknowledgment that the choice to work only with $l_\infty$-norm attackers is a limitation of your evaluation. Although it is a popular attacker, $l_2$ and $l_1$ norms have also been adopted in the literature.
> > - The discussion on polynomial verifiability in the related work section, as you mentioned in your response.

---

> > > ### Author Response · Authors · 2024-08-09
> > >
> > > Thank you for your response. We will absolutely add both points that you requested in the next version of the paper!

---

### Official Review · Reviewer_hmU3 · 2024-07-10

**Soundness:** 3
**Presentation:** 2
**Contribution:** 2
**Rating:** 5
**Confidence:** 4

**Summary:**

This paper studies adversarial attacks on tree ensemble models. The main contribution of this paper is to speed up (compared to *kantchelian* and *veritas*, please let me know if there is any misunderstanding) the process of crafting adversarial examples of tree ensemble models. It is claimed, under the setting of repeated adversarial attacks against the same model, that adversarial examples for tree ensembles tend to perturb a consistent but relatively small set of features, which is an interesting phenomenon. This paper advances the understanding of the adversarial robustness of tree ensemble models.

**Strengths:**

+ As its title suggests, this paper proposes a faster method (compared to *kantchelian* and *veritas*) for performing repeated adversarial attacks, which would help the development of more robust tree ensemble models. The claim is backed by comprehensive experiments.

+ The discussion in section 3 is easy to understand.

**Weaknesses:**

+ This paper does not comprehensively discuss the **related works**, which would lead to several problems:
    +  To my knowledge, neural network is the most popular model in modern machine learning. I suggest further comparing the tree-ensemble models and neural networks in section 5. Besides, the existence of adversarial examples is first discovered in image classification (using NNs). I suggest providing an example of adversarial examples (in real-world applications) for tree ensemble models. I noticed that some experiments are on vision datasets like MNIST and FMNIST. I wonder whether tree ensemble models can beat (deep) neural networks in some tasks.
    + This paper only mentions two adversarial attack methods for tree ensemble methods. I cannot properly evaluate the contribution of this paper without a comprehensive comparison with related works.

+ As mentioned in the "Questions" part, the presentation of this paper can be further improved.
+ It is confusing to use lowercase Italian letters to represent algorithms and datasets.

**Questions:**

+ In Line 20, "generating adversarial examples is an NP-hard problem" seems unreasonable since many existing methods can efficiently craft adversarial examples. Do you mean generating adversarial examples for tree ensemble is an NP-hard problem?
+ In lines 28-29, the authors mentioned that previous works treat the clean examples in isolation while they make use of the regularities between clean examples to speed up adversarial attacks. The term "regularity" many times in this paper, but what is this regularity and what is the intuition behind this idea? I think it is better to illustrate the idea of "regularities between clean examples" with a figure.
+ In Lines 48-48, "we propose a theoretically grounded manner to quickly find this set of features". I cannot find the results corresponding to this claim.
+ What is the meaning of SAT and UNSAT? I suggest explaining the abbreviations when they first occur.

**Limitations:**

There is a limitation statement in Lines 312-313.

---

> ### Author Rebuttal · Authors · 2024-08-07
>
> **NNs vs Tree Ensembles**
> Just because "neural network is the most popular model in modern machine learning" does not imply that all research should solely focus on them; we believe that diversity is also important.
> Tree ensemble models like XGBoost are extremely popular, very easy to apply and still outperform NNs on tabular data (Grinsztajn et al. 2022).
>
> We will add some references to techniques for the evasion of neural networks in Section 5. However, most state-of-the-art algorithms are now tailored to a specific model type. That is, they exploit properties of the model, e.g., the Kantchelian's MILP encoding exploits the logical structure of a decision tree. Hence, there are not always strong parallels between approaches for NNs and decision trees.
>
> **Examples of adv. ex. for tree ensembles**
> Adversarial examples are defined for tabular data as well as for image classification, and can therefore be generated for both NNs and tree ensembles. Consider a bank deploying a model to accept or deny loan requests by customers. A hacker may imperceptibly alter some user’s sensitive data to force a different outcome, e.g. a loan is rejected... but is accepted by subtly adding one month to the customer’s work seniority. We will add the example in the paper.
>
> **Not enough baselines**
> The reviewer does not provide any concrete baselines that they would expect to be included in the analysis.
> As an exact attack, we used *kantchelian*'s implementation as it proved to be faster than alternative SMT encodings (cites 9-12). We then considered several state-of-the-art approximate evasion attacks. We choose *veritas* as it is the best performing one to the best of our knowledge. In Section B.1 in the Supplement, we show how *veritas* is preferable with respect to another popular evasion method (LT attack). Thus we are unsure what the reviewer expects without specific references.
>
> **Problem Hardness**
> In line 20, we mean that Kantchelian et al., ICML 2016 (cite 20 in the paper) showed that determining whether an adversarial example exists with a certain distance epsilon is NP-complete for decision trees. Finding the nearest adversarial example for a given normal example is not a decision problem, and hence we used the term NP-hard.
> The same problem has been shown to be NP-complete for other model classes such as NNs [1].
>
> [1] Guy Katz, Clark Barrett, David L Dill, Kyle Julian, Mykel J Kochenderfer, Reluplex: An Efficient SMT Solver for Verifying Deep Neural Networks, in Computer Aided Verification: 29th International Conference, CAV 2017.
>
> **Regularities** The term "regularity" refers to the fact that in many attacks (for different test examples) the same small number of features are perturbed to generate a valid adversarial example. This is also shown in Figure 1. We will make this clearer in the text.
>
> **Theoretically grounded manner to identify relevant features** Section 3.2 presents an algorithm to identify a set of relevant features.
> The adoption of a statistical test ensures that the extracted feature set guarantees a low enough false negative rate on the *mixed* setting.
> Namely, the statistical test guarantees with a $1 - \eta$ probability that the false negative rate is below a threshold $\tau$.  Both $\eta$ and $\tau$ are set by the user.
>
> **SAT/UNSAT** The first place that SAT/UNSAT are mentioned is Line 76, which describes the three possible outcomes of an evasion attack. SAT/UNSAT are defined immediately afterwards in Lines 78-80: the attack is SAT if a valid adversarial example is generated, UNSAT if a solution does not exist, or TIMEOUT if a solution could not be found within the allowed time limit. These are commonly used abbreviations
> originating from mathematical logic.

---

> > ### Comment · Reviewer_hmU3 · 2024-08-09
> >
> > Dear authors,
> >
> > I have read the rebuttal, which addresses some of my concerns. I still have some follow-up questions.
> >
> > 1. About NNs v.s. Tree Ensembles. **The authors seem to have not fully understood my main concern.**
> > To be clear, I am not saying "all research should solely focus on NNs". As mentioned in my review,  adversarial examples are first observed in *vision data* (or more precisely, in the last five years, the term "adversarial examples" in the ML community often refers to vision adversarial examples first observed in 2013). Since the tree ensemble models are "very easy to apply and still outperform NNs on *tabular data*" as claimed by the authors, **there is a gap between tree ensemble models and adversarial examples.**
> >
> >     The hack's example in the rebuttal helped me understand the adversarial examples in tabular data. I suggest the authors also explain **what the adversarial examples are in Table 1.**  Providing some examples of the data and the corresponding adversarial examples in the datasets would help.
> >
> >     I also suggest the authors reconsider the choice of Figure 4. Maybe tabular data is more suitable here.
> >
> > 2. As for the baseline, in vision adversarial attacks, it is common to compare 10+ attack methods, while in this paper only two methods are compared. It is natural for a reader to ask whether this question (i.e., adversarial attacks against tree ensemble models) is in line with the community's taste.
> >
> > 3. About the Problem Hardness. According to the rebuttal, "generating adversarial examples is an NP-hard problem" in Line 20 is a term misuse. The question of "determining whether an adversarial example exists" discussed by Kantchelian et al., ICML 2016 and [1] is referred to as "robust verification/certification" in the literature, which is parallel to adversarial attack, adversarial defense, and adversarial training. The above two references can not support the claim that "generating adversarial examples is an NP-hard problem". The authors seem to be unfamiliar with the research on the adversarial robustness of machine learning models. In vision data, in most cases, the adversary can craft an adversarial example using a 10-step PGD. Therefore, I do not believe crafting adversarial examples is a computationally hard problem. Nevertheless, this paper significantly speeds up Kantchelian and veritas. The discussion here is mainly about the representation.

---

> > > ### Author Response · Authors · 2024-08-09
> > >
> > > We will respond to your points in reverse order.
> > >
> > > 3. Section 4.2 of Kantchelian et al. has a section entitled "Theoretical Hardness of Evasion".
> > >
> > > This section studies the computational complexity of constructing what they call an evading example, which is what we refer to as an adversarial example for tree ensembles. The proof in this section of their paper shows that it is possible to cast a known NP-complete problem (3SAT) as an instance of an evasion problem for tree ensembles using a linear reduction. This means the usual: if one had a polynomial time evasion algorithm, it would be possible to solve any problem instance of the well-known NP-complete problem in poly time, which is generally seen as a contradiction.
> > >
> > > Thus Kantchelian et al. is indeed showing what we state in the paper wrt to the hardness of constructing adversarial examples for tree ensembles.  We would also highlight that other tree-based evasion attack papers make similar statements about the hardness of the problem (e.g., https://arxiv.org/pdf/2010.11598).
> > >
> > > This theoretical result is not at odds with statement that in practice it may often be possible to easily find evading/adversarial examples; just like one can efficiently solve many 3SAT problems using modern satisfiability solvers.
> > >
> > > 2. We agree that is absolutely OK to ask to include additional baselines in an experiment. However,  "I cannot properly evaluate the contribution of this paper without a comprehensive comparison with related works" is neither a constructive nor actionable comment. Could you please provide a specific attack that you would expect to see included in our evaluation?
> > >
> > > While there may be 10s of approaches for NN, there are far fewer for trees. As far as we are aware, approaches developed for NNs are not directly applicable to trees. If you have specific attacks for NNs that are applicable to trees, please let us know.
> > >
> > > As stated in our initial rebuttal, we have provided justifications for omissions in the paper. We omitted SMT-based approaches (See appendix E) because they are an exact approach like Kantchelian but do not perform as well as Kantchelian.  Similarly, as discussed in B.1 we have omitted the LT attack because it has the same success rate while being an order of magnitude (or more) slower than Veritas.  We are not convinced that it is meaningful to try to improve on the less performant approaches: even with improved run times one would still prefer the more performant Kantchelian and Veritas approaches in practice.
> > >
> > > 1. We will include the hack example and further explain adversarial examples in Tabular data. For another example of this in tabular data is about domain name registrations which is discussed in the introduction and expanded upon in our response to reviewer SynL.

---

> > > > ### Comment · Reviewer_hmU3 · 2024-08-12
> > > >
> > > > Dear authors,
> > > >
> > > > Thanks for your quick reply, which answers all my questions. I suggest revising the paper according to the reply in future revisions.
> > > >
> > > > In summary, this paper studies an interesting problem about speeding up the adversarial attack against the tree ensemble models. Although this problem does not attract much attention from the community, it helps improve the performance of adversarial attacks in some specific scenarios. I will raise the score to 5.

---

> > > > > ### Author Response · Authors · 2024-08-13
> > > > >
> > > > > Thank for you incorporating our responses into your assessment of this work. We will indeed revise the paper for its next iteration.

---

### Official Review · Reviewer_s5LR · 2024-07-11

**Soundness:** 4
**Presentation:** 4
**Contribution:** 2
**Rating:** 7
**Confidence:** 2

**Summary:**

This paper proposes a new mechanism for performing computationally-efficient adversarial attacks on decision tree ensembles. Specifically, it considers the setting where an attacker wants to attack *many* samples in a dataset at the same time, and considers the *average* time to attack each sample. (The paper focuses specifically on the L_infinity threat model, but could be applied more generally.)

The proposed method is inspired by the empirical observation that, when attacking various different samples on the same trained model, attacks on decision tree ensembles tend to focus on only a subset of the total space of features, and consistently leave many features un-perturbed for all samples. The key idea is to limit the search space of the attack on later samples to only those features which are likely to be necessary to perturb, based on earlier attacks. This speeds up the attacks on the later samples.

This paper therefore proposes an algorithm that, after fully attacking only a small number of samples (using a "base" attack algorithm that can either be a full-verification MILP solver or a heuristic technique) identifies the features most likely to be involved in an attack; specifically, it ranks all of the features by how often they are perturbed. Then, using several additional small subsets of samples, the algorithm determines, coarsely, the minimum number of features that must be considered in the search space (i.e., the cutoff in the ranking) in order for the attack success rate, perturbing only these features, to be close, with high probability, to the attack success rate when perturbing all features. Finally, the attacker attacks the rest of the samples, only perturbing the identified subset of features.

The paper considers two attack variants: the "pruned" variant which, in the final stage, *only* performs the attack on the subset of features, and the "mixed" variant, which, if the attack fails in the specified L_infinity ball when considering only the subset of features, will then "fall back" on the full-feature-space search attack.  The "mixed" variant is guaranteed to have the same success rate as the full search, so only time comparisons are relevant when evaluating its success. Over a wide variety of standard datasets and decision tree ensemble models, the proposed methods are shown to produce significant speedups in attacks.

**Strengths:**

- The presentation is extremely clear and precise, and the paper appears to be very technically sound.
- The problem setting is interesting, and the results are compelling.
- Assuming that this is in fact the first paper, as claimed, to consider the "high-throughput attack" setting for adversarial attacks on decision tree ensembles, then it seems to be a highly impactful result. (However, this is not my area of expertise, so there may be prior work that I am not aware of.)

**Weaknesses:**

- The scope of interest in this work is perhaps somewhat limited:  it is fairly specific to the problem of attacking many samples on a decision tree ensemble in a batch.
- The algorithm itself is simple, and empirically (rather than theoretically) motivated. However, it appears to be highly effective, so this is not a necessarily a problem.
- The Limitations section could use some work, or be omitted: it does not mention any limitations.

**Questions:**

I have two suggestions for improvements to the algorithm; perhaps you could try these?
1. It seems odd to use effectively a "sparse" (L_0 constrained) adversarial attack when attacking under the the L_infinity threat model. Have you considered randomly perturbing (or perhaps maximally perturbing), within the L_infinity ball, all of the "other" features, that are not part of the identified sensitive-feature subset? This may give a somewhat-higher access rate on some classifiers, to the extent that the classifier is sensitive to random noise, without significantly increasing attack time.
2. In the "mixed" setting defined in the paper, the entire objective of feature pruning is to reduce runtime: there is no trade-off between final success rate and runtime. Therefore, it occurs to me that during the ExpandFeatureSet loop in algorithm 2, rather than selecting the smallest feature set that is below an arbitrary FNR threshold, one could instead just select the subset size that directly minimizes the average runtime, in the "mixed" case with fall-backs to full attacks. Note that we are already running full attacks and subset attacks on these samples, so this shouldn't take any additional time. For the sake of sample-efficiency, one elegant way to approach this would be as a stochastic multi-arm bandit problem, where the "arms" are the choices of feature set sizes, and the "reward" is the negative runtime. Any existing bandit algorithm could then be applied.

**Limitations:**

The limitations section is perfunctory, and could be expanded: for example, the limits on the situations in which this attack is relevant could be discussed further. Societal impacts are addressed.

---

> ### Author Rebuttal · Authors · 2024-08-07
>
> Thank you for the two extremely interesting and insightful suggestions! We are in the process of exploring these and we will mention these possible variations in the final version of the paper. We agree with the reviewer and we think they could be effective in some specific use cases of our algorithm.
>
> **Randomly/maximally perturbing non-relevant features**
> This is definitely a great observation! We believe that this is interesting and we are working on implementing this procedure.  Perturbing the non-relevant features should be tried when the pruned setting returns UNSAT or TIMEOUT. In these cases, the results of the pruned setting are unclear; therefore randomly perturbing non-selected features might result in a label flip and hence save the time of running a full search (if the pruned setting returns SAT, perturbing other features would not make sense: we have a valid adversarial example and changing other features could affect its predicted label). In these cases, as opposed to starting from the base example x (which would be like CUBE attack (Andriushchenko et al. NeurIPS'19)), one probably wants to start from something found by the search process.
>
> A side effect of adding random/maximal allowed perturbations is that the generated adversarial examples will be quite different than what would be found by the base algorithms (which is not problematic).
>
> More generally, we have noticed that while *veritas* uses l-inf it tends, even in the *full* search,  to apply very sparse changes. Our intuition is that this is due to the fact that we are working with trees: each split only uses one attribute and trees have limited depth which means that going from the root to the leaf only involves a small number of attributes.
>
> **Just minimize runtime rather than selecting the smallest feature set**
> This is also an extremely interesting point! The goal of the selected feature subset is indeed to reduce run time. The procedure proposed by the reviewer is effective and could benefit the *mixed* scenario. Our experiments hinted that a smaller feature subset is always reflected in a faster search, therefore we expect differences between the two approaches not to be consistent. However, this is a great idea and we are working to add it in the final version of the paper.
> Finally, we highlight that if the user has specific needs in terms of feature selection, a customized feature selection strategy can be applied in place of the one described in 3.2, and the rest of the method can still be employed as it is.
>
> **Limitations**
> We will refine the *Limitations* sections to highlight that our approach is tailored to *repeated* evasion, and not cases where a single attack is performed.

---

> > ### Comment · Reviewer_s5LR · 2024-08-13
> > **Response to Rebuttal**
> >
> > Thank you for considering my suggestions, and agreeing to expand the limitation section. I am keeping my original score of 'Accept'.
> >
> > I was slightly confused by your comment that "Our experiments hinted that a smaller feature subset is always reflected in a faster search, therefore we expect differences between the two approaches not to be consistent." To clarify, in my suggestion, I meant to refer minimizing to the _average_ runtime, _including the time for the full-feature search in instances where the "mixed" strategy falls back to performing a full search._ Therefore, a smaller feature set should not always lead to a faster average search. (Trivially, for example, if the feature set is of size one, then nearly every instance will fall back to full search, so the total time will be approximately equal to or even slightly greater than simply performing a full search.) By minimizing total average runtime directly, this should eliminate the need for an arbitrary success threshold hyper-parameter, at least in this "mixed" case.

---

> > > ### Author Response · Authors · 2024-08-14
> > >
> > > Thank you for your response! We will expand the next version of the paper mentioning the suggested variations.
> > >
> > > As for the last point, we meant “a smaller feature subset“ that still guarantees the bound on the FNR. Therefore the mentioned scenario with a feature set of size one would not be possible, as it would produce too many false negatives. The proposed alternative is now clear to us and we are working on it!

---

### Official Review · Reviewer_SynL · 2024-07-24

**Soundness:** 3
**Presentation:** 3
**Contribution:** 2
**Rating:** 4
**Confidence:** 4

**Summary:**

This paper proposes a new method for a faster generation of adversarial attacks on tree ensembles such as XGBoost and random forest. The proposed approach has two parts: first, a subset of relevant attributes in a tree is identified. This step is inspired by an empirical. observation that on tree ensembles, most of the adversarial examples tend to perturb only a limited subset of features. Once this subset is identified, the tree is pruned accordingly and the adversarial example for this pruned tree is generated. Since generating adversarial examples for a pruned tree would require less computational time, this method could generate an adversarial example faster. Theoretically, it is shown that if an adversarial example for the pruned tree is generated, it is also going to be an adversarial example for the full tree. Experimental results on a few tabular + image data shows that the proposed method could provide around an order of magnitude speed-up in generating an adversarial example.

**Strengths:**

- The paper is well-written. It guides the reader well, and provides intuitions about the theoretical aspect of the work.

- The proposed method is simple and easy to implement.

- The experimental results demonstrate considerable speed-up in generating adversarial attacks.

**Weaknesses:**

- The main weakness of this paper for me is its settings. As it is mentioned in Line 106, the threat model in the paper assumes that we have access to a subset of test examples during attack generation. This assumption plays a crucial role in the proposed method, as only by having a subset of test examples we can determine the attribute subset which is subsequently going to be used for pruning the tree. In most of the adversarial example generation literature, the assumption is that the attacker can attack the model with even a single example. Considering this, it would cast a doubt on the contributions of the paper.

**Questions:**

- Can we generate the adversarial example with only a single example?

- A figure that shows the attack success rate (y-axis) vs the number of parallel test examples (x-axis) used for attack generation would probably provide a better insight on this shortcoming.

**Limitations:**

- I would probably encourage the authors to discuss the above limitation in their paper as well.

---

> ### Author Rebuttal · Authors · 2024-08-07
>
> **Assume access to a subset of test examples during attack**
> We specifically look at the case where somebody wants to generate **many** adversarial examples.
> There are several scenarios where this is the case. In areas like phishing (or fake webshops), attackers need to generate and register many web domains. Registrars are employing automated techniques (see cite 23) to flag suspicious registrations. In this case, the attacker would know what has and has not been approved by a registrar, hence having access to unseen test examples is usually not an issue. This scenario is briefly mentioned in the intro, but we can expand upon it.
>
> Moreover, during model evaluation (i.e. before deployment) all existing methods for robustness checking require generating (or failing to generate) many adversarial examples, i.e., one for each example in a test set.  For example, this is the case for adversarial accuracy (Andriushchenko et al. NeurIPS'19, Calzavara et al. DMKD'20, Vos \& Verwer ICML'21) and empirical robustness (Kantchelian et al. ICML'16, Chen et al. NeurIPS'19, Devos et al. ICML'21). Hence, in these relevant practical scenarios our approach would provide consistent run time improvements.
>
> **Can we generate the adversarial example with only a single example**
> Yes because the method would just fall back to the original search procedure (i.e., Kantchelian or Veritas) on the full ensemble. However, this would not be an interesting use case for our approach and there would be no benefit to using it to just generate one example.
>
> **Figure with "number of parallel test examples" vs attacks success rate**
> We are not sure we correctly interpret the reviewer's suggestion. If the "number of parallel test examples" stands for the number of examples used for the feature selection procedure described in 3.2, this number varies at each run depending on the results of the statistical test: as soon as it is guaranteed that the extracted feature subset ensures a low enough FNR, the procedure terminates, as outlined in Algorithm 2. Therefore in some datasets 100 examples are enough, and in some other cases we need up to 500 examples, making it difficult to summarize everything in a single figure. If the reviewer has a specific suggestion on how to do so, we will definitely insert it in the final version of the paper.

---

> > ### Comment · Area_Chair_c2UA · 2024-08-12
> >
> > Dear Reviewer SynL,
> >
> > Can you please comment and indicate if your questions/concerns have been addressed by the authors' response?  If you have any follow-up clarification questions, please ask your questions _as soon as possible_.  We do not want to leave less than 24 hours to the authors to comment/respond to your response.
> >
> > Your AC

---

### Official Review · Reviewer_pRup · 2024-07-25

**Soundness:** 3
**Presentation:** 3
**Contribution:** 2
**Rating:** 6
**Confidence:** 3

**Summary:**

The main goal of this paper is to show that it is feasible to speed up the generation of adversarial examples against tree ensembles by perturbing only a small subset of the feature space. Their approach expands the current literature by offering an algorithmic alternative that takes advantage of the smaller subset of features instead of generating new examples from scratch. The paper shows empirical evidence from experiments on numerous datasets across different domains to show the speedup in the generation of such adversarial examples.
The key to the approach is to use a mixed approach to generating examples, where the algorithm uses the only the subset of features, or pruned setting, and only if it fails to deliver a satisfactory result or times out do they fall back to using the full method, which will find an example if one exists. Naturally, one would ask how to determine such a subset of features, which the authors propose a statistical test to add features that minimize the false negative rate by a given threshold.
Experimentally, they tested three tree ensembles on 10 datasets that were coerced to binary classification problems. The goal was to see how quickly they could generate 10,000 adversarial examples using their proposed methodology vs the current standard full approach. Evidence suggests that by pruning the ensemble to a subset of features significantly improves the generation of adversarial examples as well as helps scale the solution as tree depth increases when compared to the full approach.

**Strengths:**

The biggest strength of this paper is that it is clearly written and easy to follow for a reader that has experience in tree-based methods for machine learning. Claims are well supported by experiments and there appears to be no significant roadblocks to reproducibility given the open nature of the datasets and that the authors plan to release code along with paper acceptance. Results are reported in both summary and detailed formats. The flow of the paper is consistent with current norms for papers in this vein of research.
Given that computational resources for large scale ML projects are at a premium, results like this can reduce the time to accomplish common tasks such as adversarial example generation are very helpful to practitioners and other researchers. If methods like this are then implemented in common packages, they can be of significant impact to the community writ large.

**Weaknesses:**

The largest weaknesses in this paper, to me, are around its originality and overall theoretical impact on the field. This is an incremental improvement to adversarial generation that has its theoretical underpinnings in feature selection of trees and tree ensembles. There is significant prior work on feature selection for trees, and although this is a novel implementation of this concept for adversarial example generation speedup, it is not a concept that is particularly novel. The paper's reliance on a statistical test for feature selection introduces an element of sensitivity to parameter choices, such as the threshold for the test and the timeouts for the pruned and full methods. A detailed sensitivity analysis could help in understanding the robustness of the method to these parameter settings.

**Questions:**

Do you have any data on parameter sensitivity? How sensitive is the method to the choice of parameters, such as the threshold for the statistical test and the timeouts?
How do different dataset characteristics, such as feature correlation and data sparsity, affect the performance of the proposed method? Are there specific types of datasets where the method performs particularly well or poorly?

**Limitations:**

There are limitations on the approach’s effectiveness given different dataset characteristics. The approach works better on high-dimensional data, but to what extent? Additionally, the authors could have added some statements of the risks of the use of this approach to create adversarial examples that could enable evasion attacks that could affect common systems. It may be helpful to explain the implications or informed hypotheses if this approach could better harden tree ensemble models and therefore reduce the threat. On another note, a limitation of this approach is the tradeoff for speed over completeness. It isn’t readily apparent to me how robust this method is to simply cherry picking the most vulnerable feature and creating all the examples from minor perturbations in that feature. So, while the generated examples are valid, they could represent a fragile set of examples.
Additionally, the datasets used are only for binary classification tasks, is this method appropriate or are the results as significant when given tasks outside binary classification?

---

> ### Author Rebuttal · Authors · 2024-08-07
>
> **Originality / Impact**
> The paper's impact and originality lie in the insight that we can efficiently perform evasion attacks by only perturbing a restricted set of features.
> To our knowledge, there is little work on exploiting the sequential nature of performing evasion attacks, i.e., existing methods solve problems in isolation and do not exploit similarities among problems. This is useful for robustness checking of learned models (e.g., computing adversarial accuracy, cf. cites 1-4-6-8-20-28 from the paper). See also the response to Reviewer SynL for other cases this may be useful.
>
> One difference with respect to feature selection is that traditionally one selects features prior to training a model, i.e., the model is learned with a reduced feature set. We identify features after the model is learned (perhaps this is more akin to feature importance) and use the found features to simplify the model.
>
> **Parameter Sensitivity**
>
> **Threshold for statistical tests**
>  This would be interesting. In general, the threshold on the FNR $\tau$ is inversely proportional to the number of chosen features: the larger the selected feature subset, the lower the FNR will be. This is in tension with the goal of using as little of a feature subset as possible, to speed up the *pruned* setting. Moreover, a larger confidence $1-\eta$ increases the number of selected features as it shrinks the confidence interval for the empirical FNR (see 3.2).
>
> We ran the sensitivity analysis proposed by the reviewer on a single dataset. The table below shows the *mixed* speed-up for *miniboone* (*veritas*, XGBoost) using different values for $\tau$ (FNR) and $1-\eta$ (confidence).
> For all parameter settings, our approach improves upon the run time compared to always running a full search. For a fixed $\tau$, varying $1-\eta$ does not impact the selected features subset or performance.  When $\tau = 0.05$, many features are selected and hence there is less pruning.  $\tau=0.1$ and $\tau=0.25$ perform identically. When $\tau=0.5$ the feature subset becomes very small, and while the *pruned* setting is extremely fast there are many UNSATs and hence more calls to the *full* search. We will add a more expansive analysis in the Supplement.
>
> | tau vs 1-eta  | 0.8   | 0.9     | 0.95
> |---------------|-------|---------|------
> | 0.05          | 1.5x  | 1.5x    | 1.5x
> | 0.1           | 2.6x  | 2.6x    | 2.6x
> | 0.25          | 2.6x  | 2.6x    | 2.6x
> | 0.5           | 1.7x  | 1.7x    | 1.7x
>
>
> Our experiments used a conservative approach and set the false negative rate $\tau$ below 0.25 with confidence $1-\eta=0.9$. Empirically, this worked well and we achieve better results than the theory guarantees (average FNR of 7.5\%, see Q3 in Section 4).
>
> **Timeouts**
> Given the hardness of evasion (Kantchelian et al. ICML'16), timeouts always need to be explicitly handled (see lines 75-80).
> The chosen timeouts of 1 (*kantchelian*) and 0.1 (*veritas*) work for all the adopted datasets. Increasing the timeout parameter will not have a big impact as timeouts are rare as shown in Table 7 in the supplement. Decreasing it will lead to too many calls to the full procedure, negating the benefits of pruning.
>
>
> **Dataset Characteristics**
>
> **How do feature correlation and data sparsity affect the performance of the proposed method?**
>  Given that we select a subset of relevant features, correlation between features is likely unproblematic as redundant features will likely not be selected by the feature selection algorithm. If a certain feature is spare, it will likely be left out from the subset of relevant features, as the signal that the model can pick up will be weaker.
>
> **Are there specific types of datasets where the method performs particularly well or poorly?**
> The method generally worked well on the diverse set of datasets made of tabular data of various dimensionality and image data we considered.
> The approach would not excel when the dataset has low dimensionality (hence it does not make sense to extract a subset of features) or has a fully categorical domain (where the l-infinity norm loses meaning).
>
> **High dimensional datasets: to what extent?**
> We verified that our approach does not bring consistent run time improvements for datasets with less than 25 features.
> Therefore, we considered all the popular high-dimensional benchmark datasets we could find in related works. Suggestions for extra datasets are always welcome!
>
>
> **Risk Statement**
> We will add a clearer risk statement. We will highlight that that while this work does make attacking tree ensembles faster, being able to efficiently perform repeated evasion attacks is a key to understand what attackers can do, and to improve the applicability of robustness checking and hardening techniques, which ultimately mean reducing the threat on deployed models.
>
>
> **Robustness against simply modifying the most vulnerable features?**
> Only modifying the most vulnerable feature would not work well in general and would indeed produce fragile examples. We identify a subset of relevant features so that on one hand we avoid losing time on "irrelevant" features, and on the other hand we still produce diverse enough adversarial examples. We discuss the quality of generated examples in Section D of the Supplement, using two different measures. In general, our method produces valid adversarial examples of sufficient quality (using *veritas*, even better than the *full* setting).
>
> **Tasks beyond binary classification** Most existing approaches to performing evasion attacks on tree ensembles are defined for binary classification, which is why this paper follows the same approach (cites 6-8-20-33).
> However, we do not see any reason why our approach should not work even on multi-class and regression problems.

---

> > ### Comment · Reviewer_pRup · 2024-08-08
> >
> > I have received the rebuttal comments and appreciate the author's response. Particularly, thank you for performing and reporting on the sensitivity analysis for the parameters. I also agree with your statements on the dataset characteristics, thank you for the clarifications. I will continue to review and discuss with the other reviewers.

---

> > > ### Author Response · Authors · 2024-08-09
> > >
> > > Thank you. We will add the clarifications and include the sensitivity study in the next version of the paper!

---

### Decision · Program_Chairs · 2024-09-25

**Decision:**

Accept (poster)

**Comment:**

The paper investigates the adversarial robustness of tree ensembles.  Along these lines the authors propose that only a small number of features are important for adversarial mispredictions.  The paper proposes an algorithm that can be used in order to identify the set of features that are vulnerable to small perturbations that could cause mispredictions of the perturbed instance.  The authors validate their proposed method using a diverse set of datasets that includes both tabular and image data.

For the largest part the reviewers were satisfied with the paper and the responses received during the rebuttal and there has been mutual agreement on certain cases for certain clarifications to be provided in the final version of the manuscript.